# Learning about the Ellsberg Paradox reduces, but does not abolish, ambiguity aversion

Ruonan Jia[1,2]*, Ellen Furlong[3], Sean Gao[2], Laurie R. Santos[4], Ifat Levy[1,2,4,5]*

**1** Interdepartmental Neuroscience Program, Yale University, New Haven, Connecticut, United States of America, **2** Department of Comparative Medicine, Yale University, New Haven, Connecticut, United States of America, **3** Department of Psychology, Illinois Wesleyan University, Bloomington, Illinois, United States of America, **4** Department of Psychology, Yale University, New Haven, Connecticut, United States of America, **5** Department of Neuroscience, Yale University, New Haven, Connecticut, United States of America

* ruonan.jia@yale.edu (RJ); ifat.levy@yale.edu (IL)

**Data Availability Statement:** All relevant data are within the manuscript and its Supporting Information files.

**Funding:** This work was partially funded by NIH (National Institutes of Health, https://www.nih.gov/

## Abstract

Ambiguity aversion–the tendency to avoid options whose outcome probabilities are unknown—is a ubiquitous phenomenon. While in some cases ambiguity aversion is an adaptive strategy, in many situations it leads to suboptimal decisions, as illustrated by the famous Ellsberg Paradox. Behavioral interventions for reducing ambiguity aversion should therefore be of substantial practical value. Here we test a simple intervention, aimed at reducing ambiguity aversion in an experimental design, where aversion to ambiguity leads to reduced earnings. Participants made a series of choices between a reference lottery with a 50% chance of winning $5, and another lottery, which offered more money, but whose outcome probability was either lower than 50% (risky lottery) or not fully known (ambiguous lottery). Similar to previous studies, participants exhibited both risk and ambiguity aversion in their choices. They then went through one of three interventions. Two groups of participants learned about the Ellsberg Paradox and their own suboptimal choices, either by actively calculating the objective winning probability of the ambiguous lotteries, or by observing these calculations. A control group learned about base-rate neglect, which was irrelevant to the task. Following the intervention, participants again made a series of choices under risk and ambiguity. Participants who learned about the Ellsberg Paradox were more tolerant of ambiguity, yet ambiguity aversion was not completely abolished. At the same time, these participants also exhibited reduced aversion to risk, suggesting inappropriate generalization of learning to an irrelevant decision domain. Our results highlight the challenge for behavioral interventions: generating a strong, yet specific, behavioral change.

## Introduction

Whether we choose a dish from a menu, a medical treatment or a retirement plan, the outcome of our choices is seldom certain. While in some cases we can estimate the likelihoods for different potential outcomes (e.g. when tossing a coin), such accurate estimates are rare. In most cases, outcome likelihoods are at least partially unknown, or "ambiguous" [1]. Many

) grant R21AG049293 and NIH/NIA (National Institute on Aging, https://www.nia.nih.gov/) grant R56AG058769 to Ifat Levy. The funders had no role in study design, data collection and analysis, decision to publish, or preparation of the manuscript.

individuals exhibit aversion to ambiguity about outcome likelihoods, especially in the realm of economic decision making [1–6]. Ellsberg's Paradox provides a classic illustration of ambiguity aversion [1]. Consider two urns—A and B—each containing a total of 100 red and black balls. The ratio of red to black balls in urn A is not known, whereas urn B contains exactly 50 red and 50 black balls. The participants are asked to first choose one of the urns, and then to bet on a color in order to win a monetary reward. Drawing a ball of the chosen color from the chosen urn will result in winning money, whereas drawing a ball of the other color will lead to zero outcome. Because participants are free to choose which color will be the winning one, the winning probability for urn A is 50%, the same as urn B. To see why, consider the case in which the participant guesses that urn A contains n red balls (and thus [100-n] black balls). Because the participant could choose either color to be the winning one, the likelihood of winning money is the average of the likelihoods of winning from red and from black, that is 50% × n/100 + 50% × (100-n)/100 = 50%, the same as the unambiguous urn. But in Ellsberg's experiment—and many subsequent studies—most participants preferred to avoid the ambiguous urn even if its reward was higher [4].

Ambiguity aversion seems to be a default cognitive bias not just in financial decisions, but also in many other decision domains. This bias has been observed in studies of consumer behavior [7], health care and medical decisions [8–10], and social interactions [11]. While aversion toward ambiguity is advantageous in some situations, such as investing in insurance to avoid potential harm, in other situations it may prevent individuals from making the optimal decisions, as exemplified by Ellsberg's experiment. Indeed, augmented ambiguity aversion has been linked to psychiatric conditions, including obsessive compulsive disorder [12], post-traumatic stress disorder [13] and anxiety [14]. Developing proper interventions to modify ambiguity aversion and guiding individuals toward advantageous behavior is therefore of both practical and clinical value.

Before we attempt to modify ambiguity attitudes in real-world situations, it is useful to examine how flexible or rigid ambiguity attitudes are in Ellsberg's simple experiment. In the particular situation created by Ellsberg, choosing the ambiguous urn is always advantageous. Explaining the Ellsberg Paradox, as we did above, should thus be sufficient for eliminating any aversion to ambiguity that participants may naturally exhibit. A successful intervention in this context is a necessary condition for using this type of intervention in situations that are more complex. If, on the other hand, ambiguity aversion persists even with conscious awareness of the optimal behavior, it will suggest that explicit instruction is not sufficient for modifying ambiguity attitudes. Here we compare choices under ambiguity before and after two types of intervention, which are modeled after interventions that were successful in previous work [15–24]. Both interventions taught participants about the irrationality of their ambiguity aversion through calculations of the objective winning probability of the ambiguous lotteries. While participants in the active calculation (AC) group performed these calculations themselves, participants in the non-active calculation (NC) group were shown the process directly. A third group of participants learned about the base-rate neglect phenomenon, which was irrelevant to the task, and served as a control.

To assess the specificity of these interventions, participants also made choices under risk—another type of uncertainty in which the action-outcome association is uncertain, but the contingency is fully known. People often show risk aversion when faced with potential rewards, preferring a high chance of a small reward to a lower chance of a larger reward, even when the expected value of the latter option is higher [6,25],. For example, a risk-averse individual would prefer getting $50 for sure to playing a lottery with 50% chance of getting $120, even though the certain option has lower expected reward ($50 vs. $60). Whereas ambiguity attitude reflects the sensitivity toward missing information, risk attitude is simply the trade-off between

the magnitude of a potential outcome and its likelihood [26,27]. These two different attitudes reflect different personal traits, and are not strongly correlated with each other across individuals [6,28–32]. Indeed, some studies exploring the neural foundation of decision making under uncertainty have further suggested that the underlying biological mechanism for decisions under ambiguous and risky conditions might involve different brain circuits [28,33]. Thus, successfully intervening on people's ambiguity attitudes should ideally not affect risk preferences when the decision context changes to risky situations.

By comparing the degree of individual ambiguity and risk attitudes before and after the intervention, we asked: (1) whether and to what extent participants could reduce ambiguity aversion by learning about the Ellsberg's Paradox through AC and NC interventions; (2) whether participants would inappropriately generalize the behavioral change to risky decisions which did not involve ambiguity; and (3) whether active learning would be more effective in changing participant's behavior.

## Method

### Participants

127 adult participants (81 female, age 18.1–55.9, *Mean* = 25.7, *SD* = 8.01) were recruited from the Yale University–New Haven community. The experiment protocol was approved by Yale University Institutional Review Board, and all study participants provided written informed consent. Data from 8 participants were excluded from analysis because they preferred a lower chance to higher chance of winning the same amount of money in risky lotteries, indicating they either did not understand the task, or preferred less compared to more money (see Risk and ambiguity task). Each participant was compensated with a fixed monetary reward of $10 for completing the experiment, plus a variable bonus ($0-$65) based on their choice in the task (see Risk and ambiguity task).

### Risk and ambiguity task

The experimental design of the risk and ambiguity task was based on the Ellsberg Paradox and a previous study design [6,34]. In each trial, participants chose between a reference risky lottery with a 50% explicit chance of winning $5 (Fig 1) and another lottery whose chance and magnitude of reward both varied in different trials. The varying lottery was either risky (known probabilities of 13%, 25%, or 38%, Fig 1D) or ambiguous (probabilities were not precisely known and ranged between 13–87%, 25–75%, or 38%-62%, Fig 1C), and offered one of five amounts ($5, $9.50, $18, $34 or $65).

**Stimuli.** The chance of winning the lottery was presented physically as a bag of red and blue poker chips. Risky bags showed the exact proportion of red and blue chips, while ambiguous bags were partially covered by a grey occluder in the middle so that participants could only see a range instead of the exact distribution (Fig 1). Participants were instructed that each color would be associated with a 'win' on half the trials and with a zero outcome on the other half, and that even though probabilities in ambiguous lotteries were unknown, they would not vary between presentations of the same bag image. The physical lottery bags with red and blue poker chips were presented to the participants before the task, and were left in the room with them during the task. Participants could freely inspect the bags at the end of the experiment to verify their contents. This design resulted in 60 unique trial types ((3 risk levels + 3 ambiguity levels) × 5 amounts × 2 colors). Each trial type was repeated three times resulting in a total of 180 trials.

**Trial structure.** In each trial, participants had 6 seconds to view the two lotteries presented side by side; the reference and varied lottery were randomly assigned to the left and

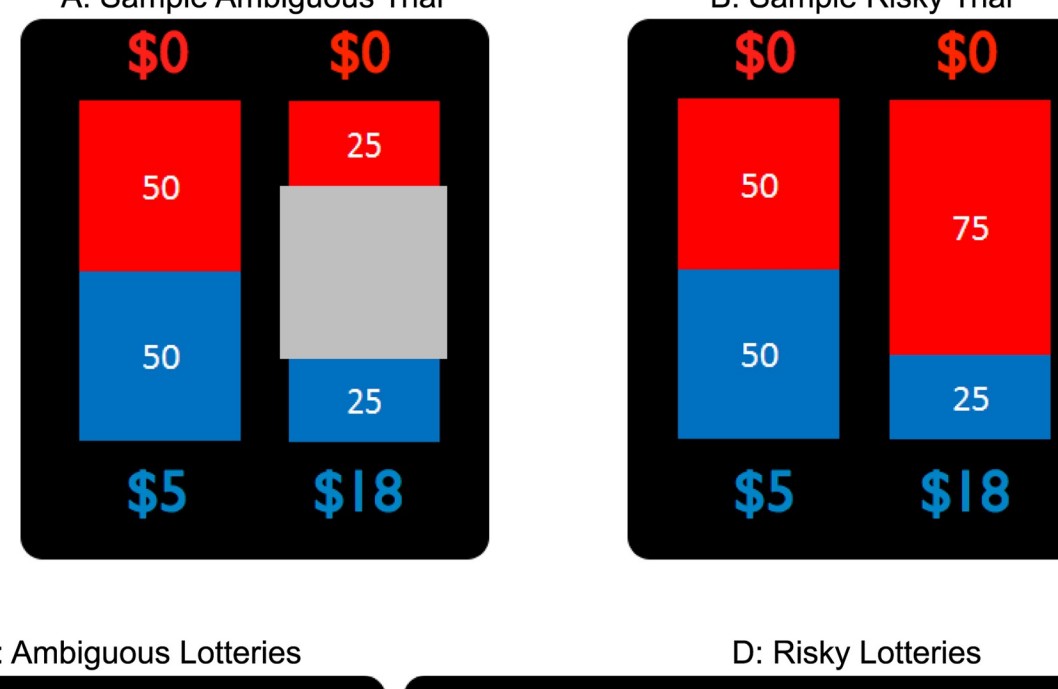

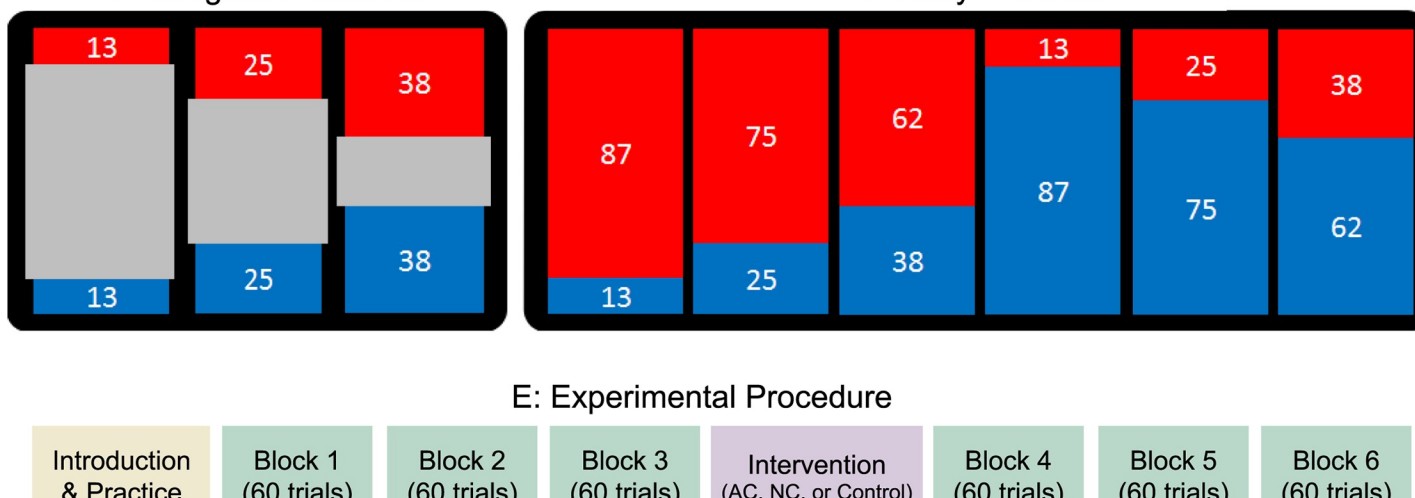

**Fig 1. Stimuli and experimental design.** (A) A sample choice between a reference lottery with a 50% chance of winning $5 (left) and an ambiguous lottery (a 25–75% chance of winning $18, right). (B) A sample choice between a reference lottery with a 50% chance of winning $5 (left) and a risky lottery (a 25% chance of winning $18, right). (C) Ambiguous and (D) risky lotteries participants encountered. The red and blue areas of the 'bag' represented the relative numbers of red and blue chips in the bag. In risky lotteries, the exact number of each color chip was known; in ambiguous lotteries, a gray occluder covered a central portion of the bag, resulting in probabilities which were not precisely known. (E) Experimental procedure. Participants first went through introduction and practice. They then completed three blocks of choices, followed by learning about the Ellsberg Paradox (AC and NC interventions) or the base-rate neglect (control) and another three blocks of choices.

right sides of the screen. Once they made up their mind, participants could respond by pressing one of two buttons, 1 or 0 on the keyboard, to choose the left or right option respectively. The pressed number (0 or 1) appeared on the screen for 0.5 second, immediately after the choice was submitted. If participants did not respond within 6 seconds, they would see a feedback of "no response". Chosen lotteries were not played during the task, so participants did not receive any feedback on the outcome of their choices, and no learning could occur. This

was done in order to ensure that the task relied solely on decision making attitudes. An inter-trial interval of 1 second followed the choice confirmation feedback, and the next trial began.

**Payment mechanism.**   Instructions encouraged participants to select their preferred lottery carefully as one randomly selected trial would be realized for bonus payment at the end of the experiment: participants would play the selected lottery by randomly picking out a poker chip out of the real physical bags, and the color of the chip determined the bonus money. Importantly, this design ensured that the objective winning probability of all of the ambiguous lotteries was 50%, the same as the reference risky lottery. To understand why, consider the ambiguous lottery with a winning chance between 25% to 75% in Fig 1A. Red was associated with the positive amount in half of the trials in which this particular image was presented, while blue was associated with the positive amount in the other half of the trials. If p was the actual probability of drawing a red chip from this bag, then (1-p) would be the probability of drawing a blue chip. Since a single trial was selected for payment at the end of the experiment, there was a 50% chance for selecting a "red-winning" lottery and a 50% for selecting a "blue-winning" lottery. The winning probability was therefore 50%×p +50%×(1-p) = 50%. Thus, regardless of the ambiguity level and of the actual distribution of blue and red chips in the bag, the winning probability of all the ambiguous lotteries was 50%. A participant aiming to maximize her earnings should therefore always choose the ambiguous lottery if it offers more than $5. Choosing the reference lottery (50% chance of $5) over an ambiguous lottery that offers more than $5 would indicate ambiguity aversion.

**Experimental procedure.**   Fig 1E shows the experimental procedure. Participants completed the experiment on a computer through E-Prime (Version 2.0), and the whole procedure lasted around 45 minutes. Participants were first introduced to the design of the task and procedure, and were shown the physical bags corresponding to the lottery pictures they saw on the computer screen. They were informed that they could freely examine the contents of the bags following the experiment. Participants then made choices in 6 practice trials, before moving to the pre-intervention phase. In this phase participants made 180 choices between risky and ambiguous lotteries, in three blocks of 60 trials each. Participants were allowed to take breaks between blocks. Following the three blocks, participants experienced one of three interventions. The (1) *active calculation* (AC; *n* = 40, 24 female, age *Mean* = 25.9, *SD* = 7.93) intervention taught participants the rationale behind the Ellsberg Paradox demonstrated in this particular study design, by walking them through the calculation of the objective winning probability of the ambiguous lottery. The calculation process engaged participants in doing the simple math themselves, and provided guidance to the correct answer if any calculation step went wrong (see S1 Text for details of AC intervention). The calculation only emphasized that the objective winning probability of the ambiguous lotteries was identical to the reference risky lottery (50%) without involving calculation of expected reward. Therefore, it reached the conclusion that participants were better off always selecting the ambiguous lottery with the larger reward regardless of how large the ambiguity level was. The (2) *non-active calculation* (NC; *n* = 40, 25 female, age *Mean* = 25.7, *SD* = 8.73) intervention was identical to AC, except that it demonstrated the whole calculation process without participants actually doing the math themselves. Finally, the (3) *base rate* (control; *n* = 39, 26 female, age *Mean* = 24.6, *SD* = 5.69) control intervention taught participants about another bias irrelevant to the task, base rate neglect [35], but still required participants to calculate probabilities (see S1 Text for details). Briefly, this intervention taught participants that base rate (the unconditional frequency of an event) should be taken into account when calculating the true probability of a specific conditional event. This was conveyed through the "blue-green cab problem": suppose a witness reported seeing a blue cab causing a hit-and-run accident; what is the probability of the witness correctly identifying the hit-and-run cab as blue, given (1) the chance of the

witness correctly identifying the color of the cab, and (2) the ratio of blue and green cabs (base rate) in the city. Similar to the AC intervention, participants were guided in performing step-by-step calculations, and reached the correct answer through feedbacks on incorrect answers. This intervention required participants to calculate the conditional probability, but was not related to ambiguity (or to value-based choice in general), and thus could serve as a control. Importantly, both the AC and NC interventions were in principle unique to ambiguity and did not apply to choices concerning two risky lotteries, but we did not explicitly mention this distinction to the participants. Note that in this design, participants did not need to perform any mental calculations while making choices following the intervention, as once they understood the Ellsberg Paradox, they should have realized they were better off always choosing the ambiguous lottery, as long as it offered more than $5. The base-rate control condition is completely irrelevant to the decision making task, and should not change participants' uncertainty attitudes after the intervention.

After the intervention, participants made another 180 choices in tree blocks, and were then asked to write a brief explanation of the Ellsberg Paradox in their own words. This last question was added to the design after six participants have already completed the study, and we therefore do not have these data for those participants. Responses were scored on a 0–3 scale, according to how many relevant features participants mentioned; specifically we scored whether participants mentioned (1) the difference between ambiguity and risk, (2) that they had an equal chance of winning in the reference and the ambiguous lottery, and (3) the need to select higher value rewards. Note that participants who underwent the control intervention were also asked to explain the Ellsberg Paradox instead of the base rate neglect, so their scores of this question should be close to zero assuming they were not already familiar with the Ellsberg Paradox.

**Participant exclusion.** We used risky trials in which participants had to choose between the reference lottery (50% chance of $5) and a lottery that offered a lower chance (13%, 25%, or 38%) of winning the same amount ($5) to test whether participants understood the task. Data from participants (n = 7) who chose the lottery with lower chance of winning $5, over 50% of the times, were excluded from the analysis, a standard procedure used in previous studies (e.g. [36]).

## Data analysis

**Estimation of risk and ambiguity attitudes.** Risk attitude: To estimate individual risk attitudes before and after intervention we examined choices in risky trials, separately before and after intervention. We only included trials in which the varying lottery offered more than $5. This is because the reference lottery (50% of $5) was clearly a better choice than the varying lottery (13%, 25%, or 38% of $5) in those trials, which were only included for verifying task understanding, and should have been chosen regardless of risk attitude. We calculated the risk attitude as the proportion of risky trials in which the participant chose the varying risky lottery instead of the reference lottery. A risk-neutral participant should choose the varying lottery over the reference if the expected value of the varying lottery is greater than that of the reference (0.5×$5 = $2.50). A risk-neutral participant should therefore choose the varying lottery on 75% of trials. A participant who chose the varying lottery less frequently was considered risk-averse, while a participant who chose the varying lottery more frequently was considered risk seeking.

Ambiguity attitude: To estimate individual ambiguity attitudes we first calculated the proportion of ambiguous trials in which the participant chose the ambiguous lottery instead of the reference lottery. Here too we did not include trials in which the ambiguous lottery offered $5,

because in such trials participants should not necessarily choose the ambiguous option, even if they have completely abolished ambiguity aversion. To make sure that this approach did not bias the results, however, we also repeated the analysis with the $5 trials included (see S2 Text).

An ambiguity-neutral participant would always prefer the varying lottery to the reference, since it offers the same chance (0.5) for a higher amount. For an ambiguity-averse participant, however, the picture is a bit more complex–her choices will rely not only on her ambiguity attitude, but also on her risk attitude. Consider a choice between an ambiguous lottery offering $10 and the reference lottery (50% of $5). While an ambiguity-neutral participant would only choose the ambiguous option in this case, an ambiguity-averse participant would first estimate the reward probability behind the grey occluder, and then apply her risk attitude to this estimated probability to calculate the expected reward of the lottery. Thus, to get a pure estimate of the participant's ambiguity attitude, we first need to account for her risk attitude. To do this, we took a model-based approach. We first fitted each participant's individual choices in the risky trials with a behavioral economics model. The subjective value of each risky option was modeled with a power utility function [25]:

$$\text{Subjective Value} = P \times V^{\alpha} \tag{1}$$

Where $P$ is the objective probability for winning the lottery (13%, 25%, 38% for risky lotteries, and 50% for the reference risky lottery), $V$ ($9.50, $18, $34 or $65 for risky lotteries, and $5 for the reference lottery) is the amount of money that the participant could win, and $\alpha$ is the individual-specific risk attitude parameter. $\alpha < 1$ indicated risk aversion, while $\alpha > 1$ indicated risk seeking.

Using maximum likelihood, the choice data in risky trials of each participant was fit to a single logistic function of the form:

$$P_V = \frac{1}{1 + e^{\gamma(SV_R - SV_V)}} \tag{2}$$

Where $P_V$ is the proportion of trials in which the participant chose the variable, rather than the reference, lottery, $SV_R$ and $SV_V$ are the subjective values of the reference and variable lotteries, respectively, calculated by Eq (1), and $\gamma$ is an individual-specific noise parameter. Risk attitude ($\alpha$) and $\gamma$ were computed separately for choice data from before and after the intervention. Risk parameter $\alpha$ was constrained within [0.1070, 2.0987], which is the range that could be detected under this task design [13].

The lower boundary was determined by equating the subjective value of the best lottery (38% chance of $65) with the subjective value of the reference lottery (50% chance of $5), using Eq 1. Similarly, to determine the upper boundary, we equated the subjective values of the worst lottery (13% chance of $9.5) and the reference lottery (50% chance of getting $5).

We then used the risk attitude ($\alpha$) and noise parameter ($\gamma$) of each participant to calculate the proportion of ambiguous choices the participant would make if she was ambiguity neutral (i.e. assumed a reward probability of 50% in all of the ambiguous lotteries). This ambiguity-neutral choice probability was subtracted from the actual choice proportion of ambiguous lotteries, to obtain the estimated ambiguity attitude, accounting for the individual's risk attitude. An ambiguity attitude of zero indicated no ambiguity aversion, and the more negative it was, the more ambiguity averse the participant was. Ambiguity attitudes were also estimated separately for pre- and post- intervention choice data.

In previous work we have modelled risk and ambiguity attitudes simultaneously with a single model [6]. However, since there was very little variability in choices under ambiguity following the intervention, we could not use the same approach here.

In addition to the model-based approach, we also examined changes in ambiguity attitude, accounting for risk attitude, in a model-free manner; details are provided in the supporting information (S3 Text).

**Mixed-effect generalized linear model (GLM) approach.** In addition to the model-based and model-free uncertainty attitudes estimation, we also conducted a mixed-effect logistic regression of choices to control for other factors and individual random effect more rigorously. We modeled the choice of the varying lottery in ambiguous and risky trials separately, using generalized linear models, with logit as the link function (Eqs 3 and 4).

$$\text{Ambiguous trials}: \text{choice} \sim \text{phase} + \text{intervention} + \text{phase} \times \text{intervention} + \text{ambiguity}$$
$$\text{level} + \text{value} + \text{EP score} + \text{EP score} \times \text{intervention} + \text{age} + \text{gender} + (1 + \text{phase}|\text{id}), \text{with} \quad (3)$$
$$\text{logit as the link function.}$$

$$\text{Risky trials}: \text{choice} \sim \text{phase} + \text{intervention} + \text{phase} \times \text{intervention} + \text{risk level} + \text{value}$$
$$+ \text{EP score} + \text{EP score} \times \text{intervention} + \text{age} + \text{gender} + (1 + \text{phase}|\text{id}), \text{with logit as the} \quad (4)$$
$$\text{link function.}$$

In addition to the primary focus of the effects of phase (pre- or post- intervention; within-subject), intervention (AC, NC, and control; between-subject), and the interaction between the two, we included the ambiguity level (only for ambiguous trials), risk level (only for risky trials), and lottery outcomes (value) as the lottery design factors that influence choices. We also included the score of understanding the Ellsberg Paradox (EP score), the interaction between this score and intervention, age, and gender as other controlling factors, and individual (id) as the random effect. Other than categorical variables including phase, intervention method, gender, and participant id, all variables were standardized before fitting the GLM. For the same reason as stated above, we excluded trials with the varying lottery offering $5 in this analysis as well. 113 participants with a score of understanding the Ellsberg Paradox were included in this analysis.

**Statistical analysis.** Participants' choice data were pre-processed to get the choice proportion of risky and ambiguous lotteries, and model-fitted in MATLAB (Version R2016b, MathWorks) to obtain the model-based ambiguity attitudes (see preprocessing and model-fitting scripts in S1 Dataset). The procedure of maximizing log likelihood (minimizing negative log likelihood) was implemented through the MATLAB function fmincon, from the Optimization Toolbox, with the aforementioned $\alpha$ constraint. Statistical analyses of $t$-test, Sign test, analysis of variance (ANOVA), post-hoc comparisons, Mann-Whitney U test, Wilcoxon signed rank test, Pearson's and Spearman's correlation were conducted by R (Version 3.5.1) [37] with packages of psych [38], ez [39], nlme [40], and emmeans [41]. All tests were two-tailed. Post-hoc multiple comparisons were adjusted with false discovery rate (FDR) controlling or Bonferroni procedures with threshold at 0.05. For GLM of logistic regression, modelling was conducted using the lme4 package [42], and test of interaction was conducted using the phia package [43].

## Results

### Manipulation check

To verify that participants in both the AC and NC groups, but not the control group, learned about the Ellsberg Paradox, we conducted a one-way ANOVA on the scores of the explanations they provided following the task (Fig 2, Table 1). This analysis revealed a group effect, *F*

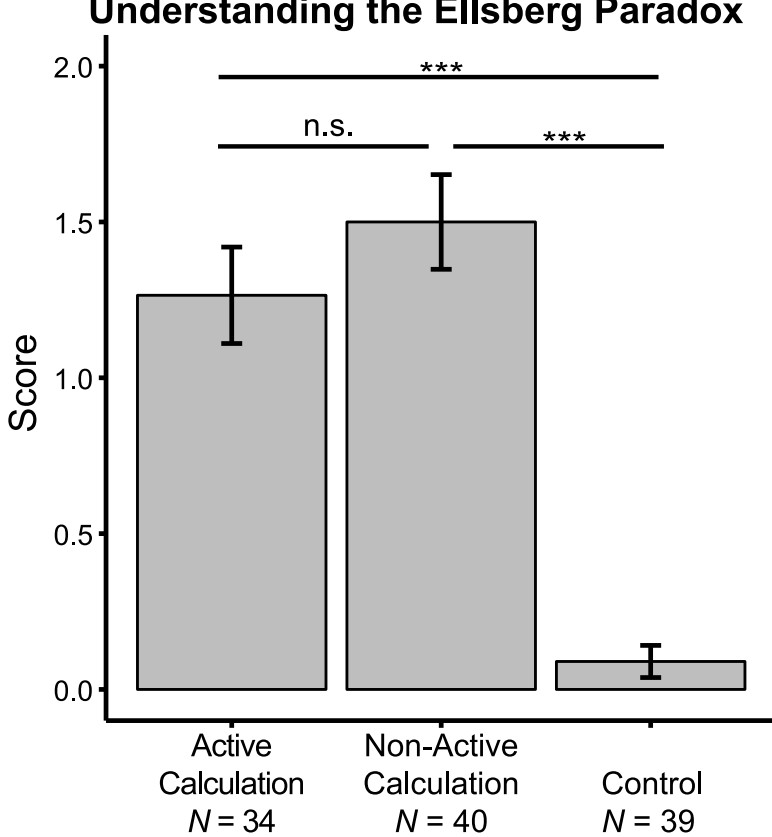

**Fig 2. Participants understood the Ellsberg Paradox after AC and NC interventions.** Mean and standard deviation of scores of understanding the Ellsberg Paradox, separately for participants from the three groups: Active Calculation, Non-active Calculation, and control. After participants finished the decision making task, they were asked to provide a written explanation of the Ellsberg Paradox. The response was scored on a 0–3 scale based on three scoring standards: whether participants mentioned (1) the difference between ambiguity and risk; (2) that they had an equal chance of winning in the reference and the ambiguous lottery; and (3) the need to select higher value rewards. Error bar represents standard error. Post-hoc $p$ values were adjusted by FDR procedure, and are labeled as: n.s., non-significant, ***, $p < 0.001$.

**Table 1. Descriptive statistics of ambiguity and risk attitudes, and score of understanding the Ellsberg Paradox for the three intervention groups, Mean (Standard deviation).**

| | Active Calculation | Non-active Calculation | Control |
|---|---|---|---|
| Number of participants | 40 | 40 | 39 |
| Ambiguity attitude (ambiguous choice proportion—modeled 50% risky choice proportion, pre-intervention) | -0.23 (0.19) | -0.22 (0.20) | -0.22 (0.20) |
| Ambiguity attitude (ambiguous choice proportion—modeled 50% risky choice proportion, post-intervention) | -0.09 (0.16) | -0.12 (0.22) | -0.27 (0.24) |
| Increase in ambiguity tolerance | 0.14 (0.21) | 0.10 (0.19) | -0.05 (0.16) |
| Risk attitude (risky choice proportion, pre-intervention) | 0.53 (0.24) | 0.51 (0.23) | 0.55 (0.23) |
| Risk attitude (risky choice proportion, post-intervention) | 0.59 (0.28) | 0.57 (0.24) | 0.54 (0.25) |
| Increase in risk tolerance | 0.06 (0.11) | 0.06 (0.16) | -0.02 (0.10) |
| Score of understanding the Ellsberg Paradox | 1.26 (0.98), $n = 34$ | 1.50 (0.96), $n = 40$ | 0.09 (0.32), $n = 39$ |

Six participants' scores of understanding the Ellsberg Paradox were not collected in the Active Calculation group.

(2, 110) = 34.1, $p < 0.001$, $\eta^2$ = 0.382; Post-hoc comparisons showed that participants understood the Ellsberg Paradox after both the AC intervention (understanding score $M$ = 1.26, $SD$ = 0.98) and the NC intervention ($M$ = 1.50, $SD$ = 0.96), and did so better than after the control intervention ($M$ = 0.09, $SD$ = 0.32; FDR adjusted $p$'s $< 0.001$). There was no significant difference between the AC and NC groups (FDR adjusted $p$ = 0.214).

## Ambiguity aversion decreased, but did not disappear, following the intervention

To estimate ambiguity attitudes, we only examined trials in which the ambiguous lottery offered more than $5. In those trials, ambiguity-neutral participants should choose the ambiguous option 100% of the time. Fig 3 presents the choices that individual participants in each group made before and after the interventions (dark grey and white respectively). Before the intervention, participants were, on average, ambiguity averse, choosing the ambiguous option on 70% of the trials ($n$ = 119, $M$ = 0.70, $SD$ = 0.21, $t(118)$ = -16.1, $p < 0.001$, one-sample $t$-test compared to 1). A one-way ANOVA of the pre-intervention choice proportion of the ambiguous lottery, with intervention method as a between-subject factor, showed no group difference (AC: $n$ = 40, $M$ = 0.70, $SD$ = 0.21; NC: $n$ = 40, $M$ = 0.70, $SD$ = 0.21; control: $n$ = 39, $M$ = 0.69, $SD$ = 0.21; $F(2,116)$ = 0.00757, $p$ = 0.992, $\eta^2$ = 0.000130).

We first examined the effect of the intervention within each group by comparing the choice proportion of the ambiguous lottery before and after the intervention by paired Sign tests. Participants chose the ambiguous lotteries more after the AC ($S$ = 30, $p < 0.001$) and NC ($S$ = 30, $p < 0.01$) interventions, but not after the control intervention ($S$ = 12, $p$ = 0.196; Bonferroni corrected for three comparisons). We then calculated the difference between the post- and pre-intervention choice proportions of the ambiguous lotteries, and compared this difference across intervention groups. A one-way ANOVA of this difference, with intervention method as a between-subject factor, revealed a significant effect ($F(2,116)$ = 12.3, $p < 0.001$, $\eta^2$ = 0.174). Post-hoc comparisons further showed that participants in the AC ($M$ = 0.15, $SD$ = 0.18) and NC ($M$ = 0.12, $SD$ = 0.18) groups both increased the proportion of choices of the ambiguous options significantly more than controls ($M$ = -0.028, $SD$ = 0.15; FDR adjusted $p$'s $< 0.001$). Non-parametric Mann-Whitney U tests of pair-wise intervention group comparison also showed that the increase of choice proportion of the ambiguous lottery was greater in AC compared to control ($U$ = 1245, $p < 0.001$), and in NC compared to control ($U$ = 1175.5, $p < 0.001$), but there was no difference between AC and NC ($U$ = 851.5, $p$ = 1; Bonferroni corrected for three comparisons).

While this result may reflect decreased ambiguity aversion, choices of ambiguous options also depend on the individual's risk attitude. To better estimate ambiguity attitudes, we employed a model-based approach to control for risk attitude (see Method section: Estimation of risk and ambiguity attitudes). After accounting for risk attitude, participants were ambiguity averse before intervention ($n$ = 119, $M$ = -0.22, $SD$ = 0.20) as revealed by a one-sample $t$-test of ambiguity attitude compared to 0 (indicating ambiguity attitude neutrality), $t(118)$ = -12.5, $p < 0.001$. A one-way ANOVA of the pre-intervention ambiguity attitude (Fig 4A, dark violin plots), with intervention method as a between-subject factor, showed no group difference in baseline ambiguity attitude (Table 1, AC: $n$ = 40, $M$ = -0.23, $SD$ = 0.19; NC: $n$ = 40, $M$ = -0.22, $SD$ = 0.20; control: $n$ = 39, $M$ = -0.22, $SD$ = 0.20; $F(2,116)$ = 0.0531, $p$ = 0.948, $\eta^2$ = 0.000915). Paired Sign tests comparing post- and pre-intervention ambiguity attitudes within groups showed that participants increased their ambiguity tolerance after the AC intervention ($S$ = 30, $p < 0.01$), and marginally after the NC intervention ($S$ = 27, $p$ = 0.071), but not after the control intervention ($S$ = 12, $p$ = 0.101; Bonferroni corrected for three comparisons). To quantify

## Distribution of Ambiguous Lottery Choice Proportion

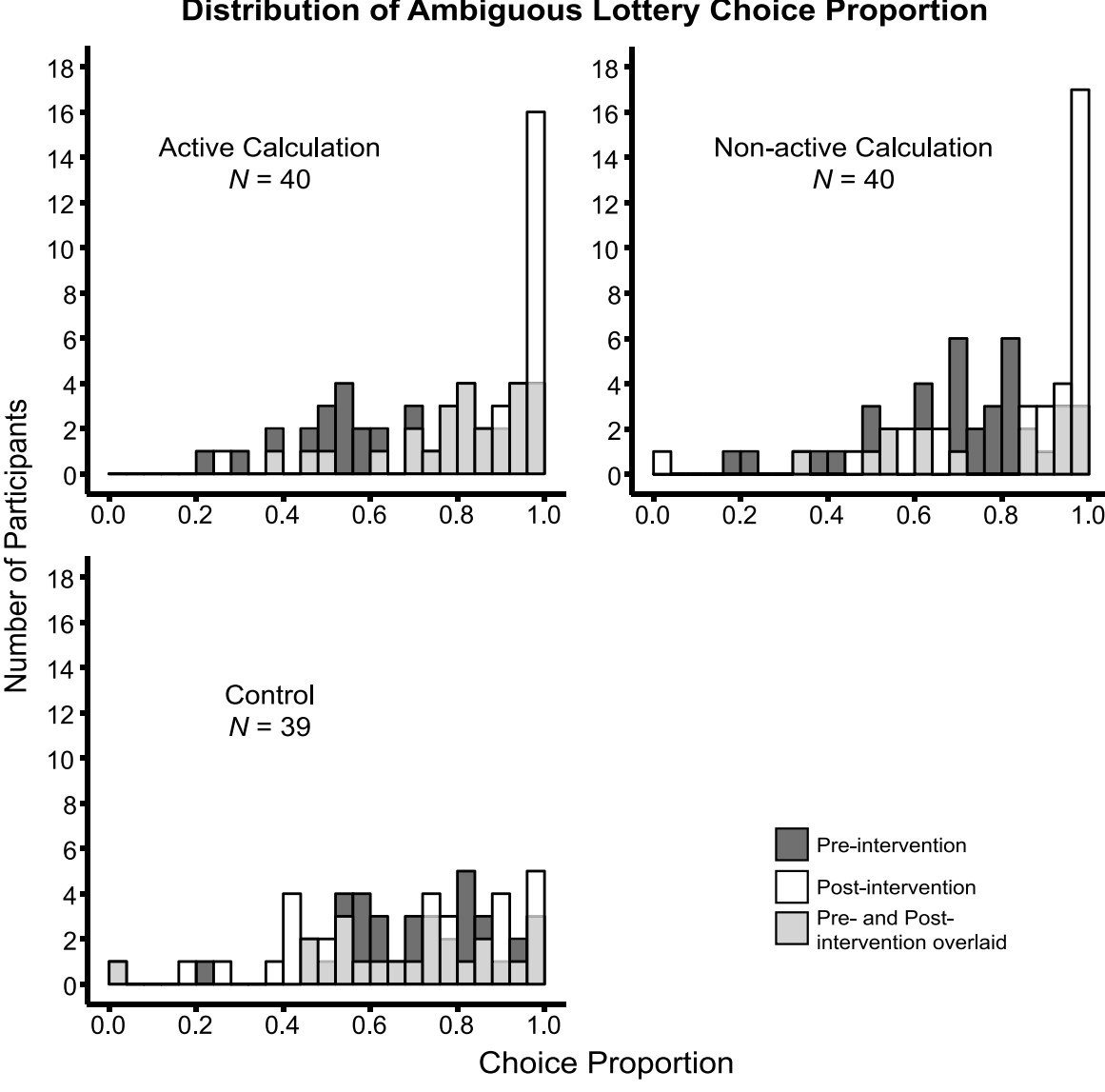

**Fig 3. Distribution of ambiguous lottery choice proportion before (dark grey) and after (white) the intervention, for each group.** The distribution of post-intervention choice proportion is overlaid on top of the pre-intervention distribution. Participants increased their choice proportion of the ambiguous lottery after the active calculation and non-active calculation interventions as shown by the shift of distributions, but not after the control intervention.

the de-biasing effect of each intervention method, we calculated the difference between the post- and pre-intervention ambiguity attitudes (i.e. the increase in risk-controlled ambiguous choices after intervention). A one-way ANOVA of this difference, with intervention method as a between-subject factor, revealed a significant effect ($F(2,116) = 11.5$, $p < 0.001$, $\eta^2 = 0.165$; Fig 4B, S1 Table). Post-hoc comparisons further showed that AC ($M = 0.14$, $SD = 0.21$) and NC ($M = 0.10$, $SD = 0.19$) interventions both decreased participants' ambiguity aversion compared with control ($M = -0.05$, $SD = 0.16$; FDR adjusted $p$'s $< 0.001$). The post-hoc comparison also showed that the difference between AC and NC was not significant (FDR adjusted $p = 0.332$), indicating that de-biasing strategies based on either active learning or passive learning performed equally well in our experiment. Non-parametric Mann-Whitney U tests of pair-

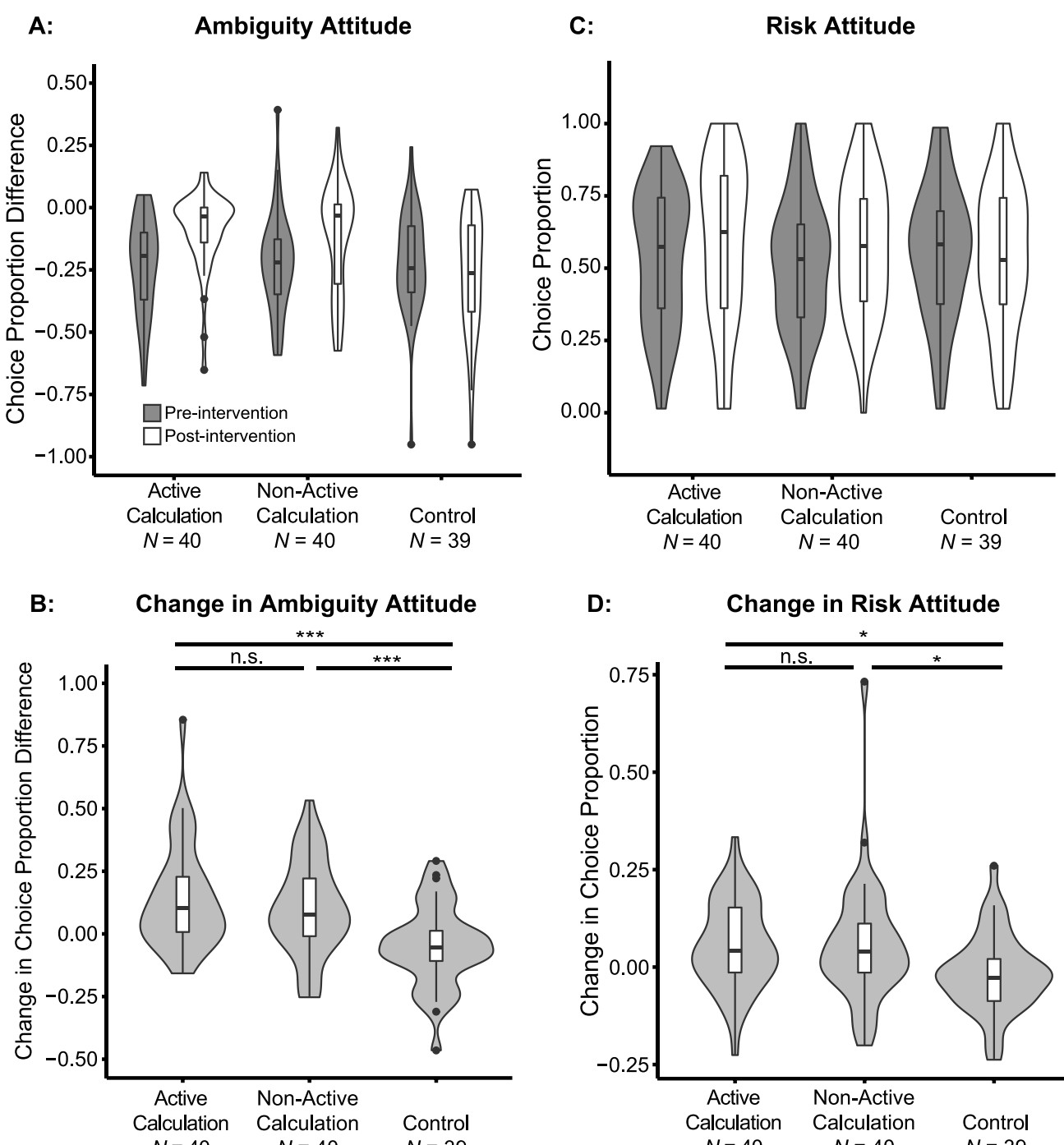

**Fig 4. Changes in ambiguity and risk attitudes following the intervention.** Violin and box plots of: (A) Ambiguity attitude, calculated as the difference between the choice proportion of ambiguous lotteries and the modeled choice proportion of 50% risky lotteries, before and after intervention for each group: Active Calculation, Non-active Calculation, and control. (B) Change in ambiguity attitude after intervention for each group. Positive number indicated increase in choice proportion, in other words, decrease of ambiguity aversion. (C) Risk attitude, calculated as the choice proportion of risky lotteries, before and after intervention for each group. (D) Change in risk attitude after intervention for each group. Positive numbers indicate increase in choice proportion (decrease of risk aversion). *P* values were adjusted by FDR procedures, and significance levels are labeled as: n.s., non-significant, $^*$, p < 0.05, $^{**}$, $p < 0.01$, $^{***}$, $p < 0.001$. Plots are trimmed within the range of the data. Box plots show the medians with horizontal thick lines. The lower and upper hinges correspond to the first and third quartiles, and the whiskers extend from the hinge to the largest value no further than 1.5 inter-qualitle range (distance between the first and third quartiles) of the data. Outliers beyond the whiskers are plotted by dots individually. Violin plots show the mirrored densities of the data.

wise group comparisons also showed the same results: the decrease in ambiguity aversion was greater in AC compared to control ($U = 1212$, $p < 0.001$), and in NC compared to control ($U = 1145.5$, $p < 0.01$), but there was no difference between AC and NC ($U = 858$, $p = 1$; Bonferroni corrected for three comparisons). For completeness, we also investigated the change of ambiguity attitude in a model-free way, by looking at choice difference between trials with different levels of ambiguity (S3 Text). This analysis has also shown that both the AC and NC interventions reduced ambiguity aversion to a similar extent.

To control for additional factors, including understanding of the Ellsberg Paradox, age, gender, and individual random effects, we modeled choices of the ambiguous lotteries using logistic regression including phase (pre- or post- intervention; within-subject) and intervention (AC, NC, and Control; between-subject) as categorical factors (see all factors included in the model in section Data analysis: Mixed-effect generalized linear model (GLM) approach). This analysis further confirmed the interaction effect of phase and intervention: participants increased choices of the ambiguous lotteries after AC intervention compared with Control ($Beta = 1.74$, $SD = 0.430$, $p < 0.001$), and after NC intervention compared with Control ($Beta = 1.77$, $SD = 0.412$, $p < 0.001$). There was no significant difference between the two interventions s ($\chi^2 = 0.00732$, $p = 0.932$). Age, and gender did not influence choices. Interestingly, understanding of the Ellsberg Paradox also did not predict change in choices of ambiguous options (see S2 Table for effects of all factors).

While the AC and NC interventions were effective, it is important to note that participants still did not choose the option with the higher reward on all trials (Fig 3, white distributions in AC and NC. AC: $M = 0.85$, $SD = 0.18$, one-sample Wilcoxon signed rank test compared to 1: $W = 0$, $p < 0.001$, Bonferroni corrected; NC: $M = 0.82$, $SD = 0.23$, $W = 0$, $p < 0.001$, Bonferroni corrected). Thus, even in the simple situation of the Ellsberg Paradox, explicit instruction did not fully eliminate ambiguity aversion.

## Risk aversion was also weakly reduced following the intervention

Ambiguity aversion was significantly reduced, but was not abolished, following the interventions. Next, we turned to examine the specificity of the effect to ambiguity. Since our interventions were specific to ambiguous probabilities, they should not have changed participants' risk preferences. To estimate risk attitudes, we simply calculated the choice proportion of the varying risky lottery among all risky trials for each participant, separately in pre- and post- intervention phase. Participants' pre-intervention risk attitude was correlated with their ambiguity attitude ($n = 119$, Pearson's $r = 0.49$, $p < 0.001$). A one-way ANOVA of the pre-intervention choice proportion of the risky lottery (Fig 4C, dark violin plots), with intervention method as a between-subject factor, showed no group difference in baseline risk attitudes (Table 1, AC: $n = 40$, $M = 0.53$, $SD = 0.24$; NC: $n = 40$, $M = 0.51$, $SD = 0.23$; control: $n = 39$, $M = 0.55$, $SD = 0.23$; $F(2, 116) = 0.363$, $p = 0.696$, $\eta^2 = 0.00622$). Paired Sign tests did not show a significant difference between the proportions of risky choices before and after the intervention in the AC ($S = 26$, $p = 0.160$), NC ($S = 24$, $p = 0.430$), or control ($S = 13$, $p = 0.219$; Bonferroni corrected for three comparisons) groups. Although the change in risk attitude was not significant according to this stringent analysis, a one-way ANOVA of the difference in risk attitude, with intervention method as a between-subject factor, revealed a significant effect ($F(2, 116) = 4.55$, $p < 0.05$, $\eta^2 = 0.0728$; Fig 4D, S3 Table). Post-hoc comparisons further showed that both AC ($M = 0.06$, $SD = 0.11$) and NC ($M = 0.06$, $SD = 0.16$) interventions increased participants' choice proportion of risk lotteries compared with control ($M = -0.02$, SD = 0.10, FDR adjusted $p$'s $< 0.05$). There was no difference between the effects of AC and NC (FDR adjusted $p = 0.971$). Non-parametric Mann-Whitney U tests of pair-wise intervention group comparison

also showed the same results: the increase of risky choice was greater in AC compared to control ($U = 1100$, $p < 0.01$), and in NC compared to control ($U = 1052$, $p < 0.05$), but there was no difference between AC and NC ($U = 834.5$, $p = 1$; Bonferroni corrected for three comparisons).

To further control for factors including understanding of the Ellsberg Paradox, age, gender, and individual random effects, we modeled choices of the varying risky lotteries using Logistic regression, including phase (pre- or post- intervention; within-subject) and intervention (AC, NC, and Control; between-subject) as categorical factors (see all factors included in the model in section Data analysis: Mixed-effect generalized linear model (GLM) approach). This analysis further confirmed the interaction effect between phase and intervention, namely that participants increased choices of the varying risky lotteries more after the AC intervention compared with Control ($Beta = 0.618$, $SD = 0.248$, $p < 0.05$), and after the NC intervention compared with Control ($Beta = 0.549$, $SD = 0.238$, $p < 0.05$). There was no significant difference between the two interventions ($\chi^2 = 0.0803$, $p = 0.777$). Among the control factors, age was negatively associated with choices of the varying risky lotteries ($Beta = -0.365$, $SD = 0.185$, $p < 0.05$), but there was no significant effect of understanding of the Ellsberg paradox or gender (see S4 Table for effects of all factors).

After the intervention, participants' risk attitude was still correlated with ambiguity attitude in the AC and NC groups (Fig 5B, $n = 80$, Pearson's $r = 0.29$, $p < 0.01$), but the correlation was marginally weaker than before the intervention (Fisher's $z = 1.77$, $p = 0.0772$). Because the Ellsberg Paradox was irrelevant to risky choices, participants should not generalize the learning to decisions involving only risky lotteries. Yet in our task, the increase of risky choices indicated that, at least to some extent, participants inappropriately generalized the learning to irrelevant decision contexts. The change in risk attitude, however, was not correlated with the change in ambiguity attitude (Fig 5C, $n = 80$, Pearson's $r = 0.055$, $p = 0.63$).

## Pre-intervention ambiguity attitude predicted the change in ambiguity aversion

If learning the Ellsberg paradox successfully decreased participants' ambiguity aversion, those who were most averse to ambiguity before the intervention should be most likely to increase their tolerance to ambiguity. To examine if this was the case, we looked at the correlation between pre-intervention ambiguity attitude (risk-corrected choice proportion in ambiguous trials) and the change in ambiguity attitude, combining participants from the AC and NC interventions. We observed a negative correlation between the two, such that participants who were most averse to ambiguity increased their ambiguity tolerance more (Fig 6A; Pearson's $r = -0.53$, $p < 0.001$). Note, however, that this negative correlation could result from a ceiling effect, as participants who were not ambiguity averse before the intervention could not reduce their ambiguity aversion any further. We did not see such relationship between pre-intervention risk attitude and change in risk attitude (Fig 6B; Pearson's $r = -0.064$, $p = 0.572$).

## Understanding the Ellsberg Paradox did not predict the change in ambiguity aversion

To look at whether participants' reduction of ambiguity attitude could be predicted by their level of understanding of the Ellsberg Paradox, we investigated the relationship between participants' score for explaining the Ellsberg Paradox and their decrease of ambiguity aversion by the AC and NC interventions. No correlation was found for either AC (Fig 7A; Spearman's $\rho = 0.035$, $p = 0.84$), or NC (Fig 7B; Spearman's $\rho = -0.17$, $p = 0.30$) interventions.

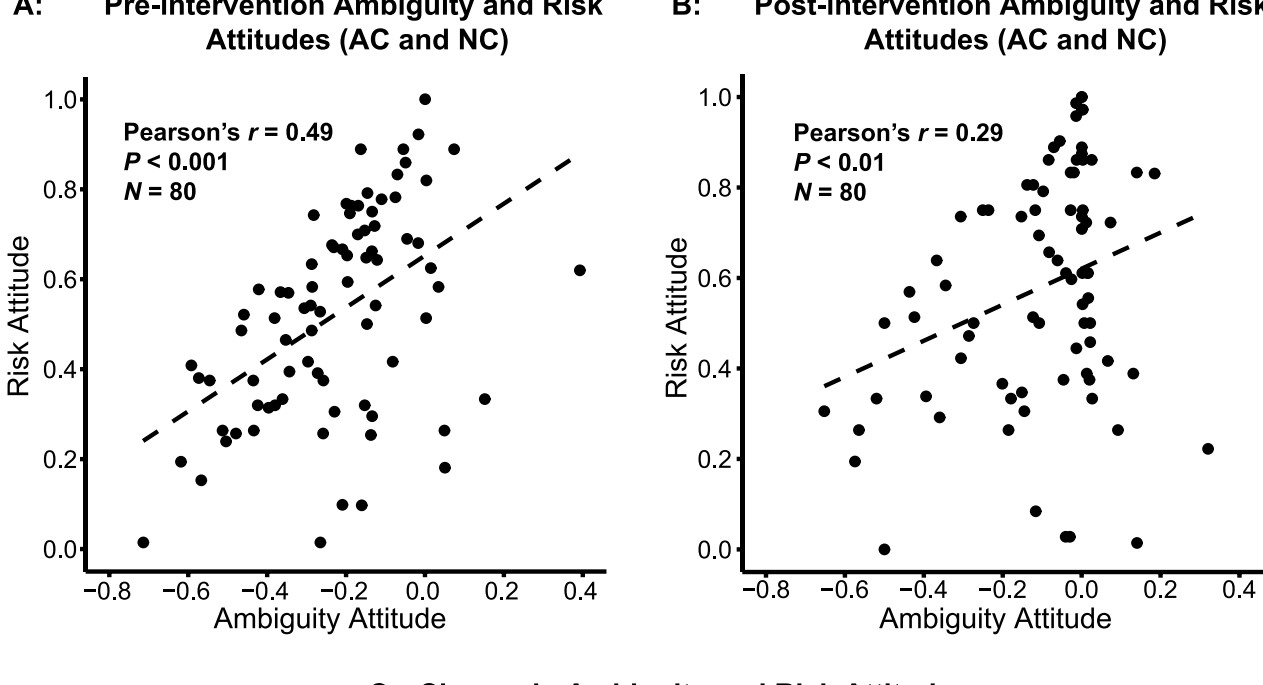

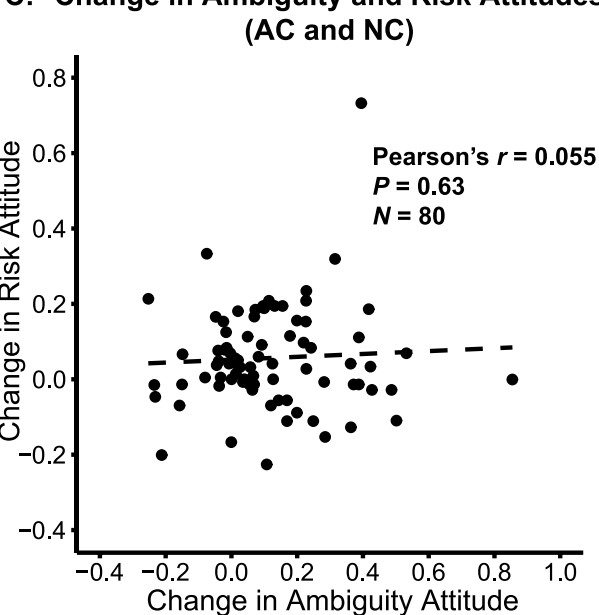

**Fig 5. Correlations between risk and ambiguity attitudes before and after the intervention, and correlation between changes in risk and ambiguity attitudes.** (A, B): The correlation between ambiguity and risk attitudes was marginally reduced by intervention among the two intervention groups: Active Calculation (AC) and Non-active Calculation (NC). Ambiguity and risk attitudes were correlated before (A) and after (B) the intervention. There was a trend for reduced correlation between risk and ambiguity attitudes after, compared to before, the intervention (Fisher's $z = 1.77$, $p = 0.0772$). (C): Change in ambiguity attitude was not correlated with change in risk attitude across AC and NC intervention groups.

### Reaction time did not reflect intervention effect

We also looked at participants' reaction time (RT; the time between onset of lottery presentation and button press). Participants were generally faster in ambiguous trials than in risky trials, in both pre- and post- intervention, regardless of the intervention methods. Faster

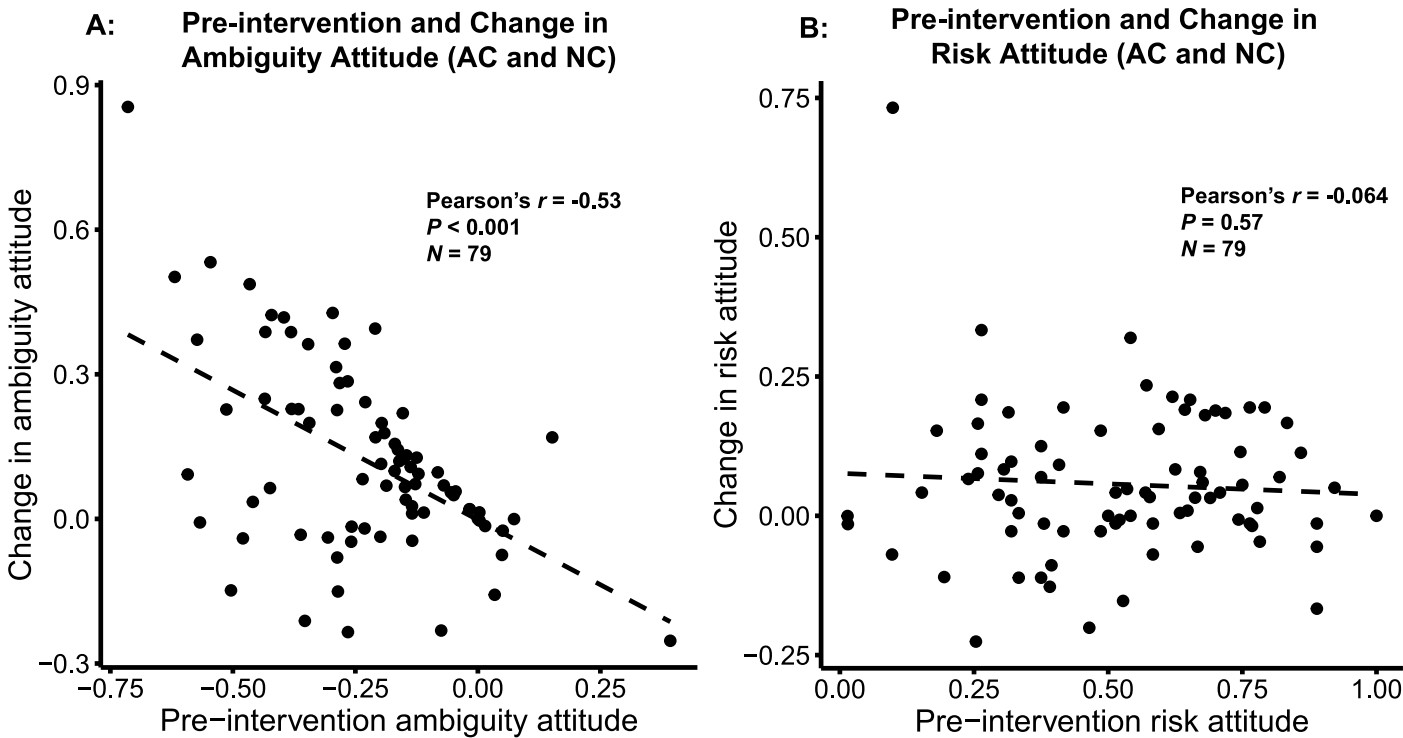

**Fig 6. Pre-intervention attitudes predict change in attitudes.** (A): Change in ambiguity attitude was negatively correlated with pre-intervention ambiguity attitude. (B): Change in risk attitude was not correlated with pre-intervention risk attitude. Correlation was calculated across the two intervention groups: Active Calculation (AC) and Non-active Calculation (NC).

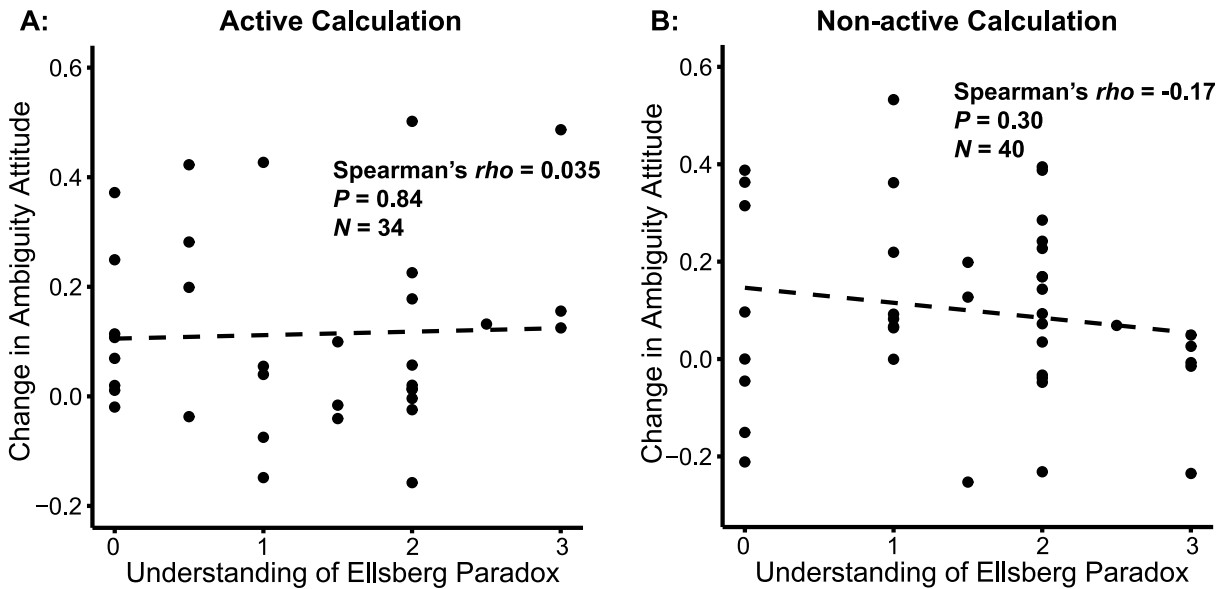

**Fig 7. Correlations between scores of understanding the Ellsberg Paradox, and the changes in risk and ambiguity attitudes.** Understanding of the Ellsberg Paradox did not predict behavioral change in ambiguity attitude: for (A) Active Calculation intervention and (B) Non-active Calculation intervention.

response in ambiguous trials suggest that participants were not conducting more calculations with the ambiguous lotteries, although they seemed more complex than the risky lotteries. RT was generally reduced after the intervention, due to familiarity with task, but there was no significant effect of intervention. Details and statistics are presented in S4 Text.

## Discussion

Ambiguity aversion is a robust phenomenon [1–6], and is sometimes maladaptive, leading to undesired results such as low stock market participation [44,45], high portfolio under-diversification [45], and social phobia [46]. In this study, we tested whether understanding the Ellsberg Paradox is sufficient for reducing ambiguity aversion, when such aversion is clearly disadvantageous. Our results suggest that awareness of the detrimental effect of ambiguity aversion in the context of the Ellsberg Paradox does reduce ambiguity aversion in this limited context. At the same time, our results point to the challenges in this kind of de-biasing. First, although ambiguity aversion was reduced, it did not disappear; most participants remained ambiguity averse to some degree. A second challenge with this de-biasing procedure is that participants also reduced their aversion to risk, suggesting inappropriate generalization of the learning to another irrelevant decision context. This is remarkable, given that we have used identical choice sets before and after the learning. Participants should have simply switched to choosing the ambiguous option when it paid more than $5, without changing any of their other choices. An effective practical intervention should generalize to situations that were not experienced during the experiment (i.e. other ambiguous situations) and at the same time not generalize to irrelevant situations (i.e. non-ambiguous situations). Thus, our results suggest that explicit instructions may have a limited effect on decision biases.

### Ambiguity aversion was reduced following the intervention

Previous studies have attempted to reduce ambiguity aversion through various intervention types. Based on the hypothesis that ambiguity aversion is caused by fear of negative evaluation [30], Trautmann et al. (2008) found that eliminating evaluations by experimenters could reduce ambiguity aversion [47]. Group decisions could also push ambiguity attitude towards neutrality [48], especially when individuals with ambiguity neutral attitudes were in the group [11]. Providing outcome feedback for each and every choice, rather than for a single choice as we did here, can also reduce ambiguity aversion [49]. These studies, however, modified ambiguity attitudes by changing the decision environment, rather than directly modifying individual attitudes in the same decision context, as we did here. In a different decision domain, Senecal et al. (2012) were successful in reducing participants' temporal discount rates, by teaching them that discount rates of delayed rewards were far higher than the market interest rate [50,51]. Our results provide evidence for the potential to modify behavior in a very different decision domain. Together, these studies demonstrate that teaching participants the theoretical approach for optimal behavior can have a substantial de-biasing effect for some individuals.

### Ambiguity aversion was not completely abolished

Unlike Knightian uncertainty [52], which cannot be resolved, in Ellsberg's paradigm (and in our design) ambiguity can be fully resolved by computing the objective winning probability of the ambiguous lotteries. Once participants learn that the objective winning probability is 50% —same as the winning probability of the reference lottery—all they need to do to maximize rewards is to choose the ambiguous lottery whenever it pays more than the reference. Still, most participants did not fully adopt this strategy, choosing the reference lottery from time to

time even after the intervention. A similar observation was made by Senecal et al (2012), whose participants did not fully reduce their discounting rate to the desired level of market interest rate, even when provided with an explicit strategy of choosing more patient options [53]. These findings suggest that, at least for some individuals, default behavioral biases go beyond a cognitive misunderstanding of the situation. A related possibility is that these individuals relied in their decisions on what the Fuzzy-Trace Theory calls the "verbatim system", rather than the "gist system" [54]. Thus, they may have acquired the details of the intervention (objective reward probability of the lottery is 50%), but not the "bottom line" (always choosing the ambiguous lottery is the best strategy).

With that said, there was considerable variation in participants' ability to modify behavior following the intervention (Figs 3 and 4B); future research will need to explore the bases for these individual differences. One such factor could be variability in cognitive ability, since understanding the Ellsberg Paradox requires understanding of probability and mathematical calculations. Indeed, many other decision biases, including overconfidence, conjunction fallacy, hindsight bias, and inefficient use of decision rules, are related to individual differences in cognitive abilities [55–57]. Although we did not directly measure cognitive ability in our participants, we did evaluate their understanding of the Ellsberg Paradox, and found no correlation between the degree of understanding and the change of ambiguity attitude. This lack of correlation was perhaps not surprising, because participants could change their behavior even if they did not fully understand the Ellsberg Paradox, but chose to adopt the recommended strategy suggested by the intervention, to always choose the ambiguous lottery. In this case, the extent to which participants were persuaded by the experimenter plays a more important role than actually understanding the rationale behind the Ellsberg Paradox. Still, cognitive factors are likely to play a role in the ability to flexibly modify behavior [58]. Another potential factor is socioeconomic status, which predicts cognitive ability during development [59]. Socioeconomic status has been linked to ambiguity attitude [60], and has been shown to affect processing of financial information [61]. Other demographic factors, such as age and life experience with uncertain situations, likely also play a role. Moreover, ambiguity aversion probably stems from multiple sources, including fear of negative evaluation [30], subjective experience of missing information [62], and comparative ignorance [63]. Decision makers may be willing to forgo financial gains in order to avoid these negative feelings; a mere understanding of the optimal strategy for value maximizing will thus not suffice to modify behavior in such individuals. Another interesting question is whether and how ambiguity aversion relates to the "optimism bias", which is commonly reported, where people actually overestimate the likelihood for positive future events (such as having a long life expectancy), and underestimate the likelihood for negative events (such as getting divorced) [64,65].

## Risk aversion was inappropriately reduced following the intervention

Following the interventions, risk aversion also decreased, although the effect size was smaller than the reduction in ambiguity aversion, and the two effects were not correlated across participants. Risk and ambiguity are separate concepts in mathematical and economical terms. In our task, risk is irreducible uncertainty, and risk attitude depends on the participant's trade-off between the magnitude and likelihood of the outcome. Ambiguity is estimation uncertainty about the outcome likelihood, and ambiguity attitude reflects the sensitivity to missing information. In real life, of course, risk and ambiguity are convolved with each other, and choices involving only risk are rare. By definition, choices made under ambiguous conditions rely both on the individual's ambiguity attitude and on their attitude towards risk. This close relationship between risk and ambiguity may explain the generalization of the learning to risky

choices. While individuals are affected differently by risk and ambiguity in their choices, they are not necessarily able to make the conscious distinction between these two types of uncertainty, and thus may treat the two in a similar manner when trying to consciously modify their behavior. In addition, our participants' attitudes towards risk and ambiguity were correlated with each other at baseline, and this correlation ($r$ = 0.49) was stronger than what has been previously reported [6,28,30,32], making it more likely that learning in one domain will generalize to the other. The generalization effect may also result from the nature of our specific interventions. As part of the learning, participants were engaged in calculating reward probabilities, and this may have shifted their attention towards further processing of probabilities in the risky trials, resulting in change of behavior. Future research should explore the similarities and differences between an explicit intervention like the one we used here, and an implicit paradigm, in which participants learn from feedback on their choices or encounter cues indicating change of context. Such research can also explore the effect of probability calculations on attention, and how attention modulations may modify uncertainty attitudes.

## Limitations of the current study and directions for future research

Our participants came from a Yale-New Haven community, among which many were Yale students or staff members. Social economic status was quite homogeneous, and was not representative of the general population. Future studies should investigate the effect of learning about the Ellsberg paradox in a broader and more diverse population.

Another potential limitation is that our interventions implicitly informed participants that their behavior had been suboptimal, which may have led to an experimenter demand effect [66], rather than true behavioral change through learning. Although we cannot completely rule out such an effect, we note that participants did seem to understand the Ellsberg Paradox (Fig 2), suggesting that the behavioral change could not be fully driven by experimenter demand. In future attempts to replicate the effect, it will be interesting to use an intervention which walks participants through probability calculations without revealing the best strategy.

In this study we demonstrate that modifying decision biases is possible, yet difficult. Future research will need to establish the potential for appropriate generalization of such interventions to different choice situations, and different domains, especially where ambiguity aversion can lead to more devastating outcomes. One such domain worth investigating is the development of psychiatric symptoms. Prior studies have demonstrated the relationship between aversion to ambiguous monetary options and clinical symptom severity, including anxiety [14], OCD [12] and PTSD [13], suggesting that economic preferences could influence, or be affected by, the development and maintenance of these disorders. Future studies could directly test whether modifying patients' ambiguity aversion in financial behavior could be generalized to alleviating psychiatric symptom, in searching for more effective behavioral treatment for psychiatric disorders. Our study only looked at the short-term effect of a single-session intervention–the difficulties we encountered even in this simple situation suggest that in order to achieve robust and generalizable effect, a multi-dimensional approach will be needed.

It will also be interesting to investigate the neural basis for the behavioral effects we observed. Overcoming ambiguity aversion requires effective cognitive inhibition over habitual behavior. Human neuroimaging studies have suggested that top-down cognitive control involves the dorsolateral and inferior PFC, frontal-parietal and cingulo-opercular networks [67–70]. Combining functional brain imaging with de-biasing tasks, it will be possible to investigate how neural circuits of cognitive control impact brain regions implicated in ambiguity processing and value-based decision making, including inferior PFC, posterior parietal cortex, amygdala, OFC, anterior and posterior cingulate cortices, vmPFC, and striatum [6,28,33,71–73].

## Supporting information

**S1 Text. Details of active calculation (AC) and non-active calculation (NC) intervention methods, and control (base rate) intervention method.**
(PDF)

**S2 Text. Analysis of ambiguity attitude including trials in which the ambiguous lottery offered $5.**
(DOCX)

**S3 Text. Model-free analysis showed that AC and NC interventions reduced ambiguity aversion.**
(DOCX)

**S4 Text. Reaction times did not differ across intervention methods.**
(DOCX)

**S1 Table. ANOVA of change in model-based ambiguity attitude after intervention.** One-way ANOVA of change in ambiguity attitude after intervention, with intervention method as between-subject factor. Ambiguity attitude was calculated as the difference between the ambiguous lottery choice proportion and the modeled 50% risky lottery choice proportion.
(DOCX)

**S2 Table. Summary of all effects from logistic regression of choices of the ambiguous lotteries.** 113 participants with a score of understanding the Ellsberg Paradox were included in this analysis. The result confirms the interaction effect between phase and interaction method, that participants in both AC and NC groups increased choice of the ambiguous lottery compared with the Control group. AC and NC groups did not show difference in this effect ($\chi^2 = 0.00732$, $p = 0.932$).
(DOCX)

**S3 Table. ANOVA of change in risk attitude after intervention.** One-way ANOVA of change in risk attitude after intervention, with intervention method as between-subject factor. Risk attitude was calculated as the choice proportion of the risky lottery in risky trials.
(DOCX)

**S4 Table. Summary of all effects from logistic regression of choices of the varying risky lotteries.** 113 participants with a score of understanding the Ellsberg Paradox were included in this analysis. The result confirms the interaction effect between phase and interaction method, that participants in both AC and NC groups increased choice of the varying risky lottery compared with the Control group. AC and NC groups did not show difference in this effect ($\chi^2 = 0.0803$, $p = 0.777$).
(DOCX)

**S1 Dataset.**
(ZIP)

## Acknowledgments

We thank Igor Berman for helping with data collection and pre-processing. The study was funded by NIH grants R21AG049293 and R56AG058769 to IL.

## Author Contributions

**Conceptualization:** Ellen Furlong, Laurie R. Santos, Ifat Levy.

**Formal analysis:** Ruonan Jia.

**Funding acquisition:** Ifat Levy.

**Investigation:** Ruonan Jia, Ellen Furlong, Sean Gao.

**Methodology:** Ruonan Jia, Ellen Furlong, Ifat Levy.

**Project administration:** Ruonan Jia.

**Supervision:** Ifat Levy.

**Visualization:** Ruonan Jia.

**Writing – original draft:** Ruonan Jia, Ifat Levy.

**Writing – review & editing:** Ruonan Jia, Ellen Furlong, Laurie R. Santos, Ifat Levy.

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
