## [Decision Letter · Decision Letter 0]

30 Sep 2019

PONE-D-19-23100

Learning about the Ellsberg Paradox reduces, but does not abolish, ambiguity aversion

PLOS ONE

Dear Ms. Jia,

Thank you for submitting your manuscript to PLOS ONE. After careful consideration, we feel that it has merit but does not fully meet PLOS ONE’s publication criteria as it currently stands. Therefore, we invite you to submit a revised version of the manuscript that addresses the points raised during the review process.

I made the decision on your manuscript after receiving comments from two scholars in the field and giving the paper a review myself.  Should you decide to submit a revision of your paper please be sure to address all the comments by the reviewers in your response document.  I would like you to in particular address the data issue raised by reviewer #1.  In addition to Rev1 and Rev2's comments please incorporate responses to a few things from me as well.

Define within the text what your "Base rate neglect" intervention taught. Many readers will not know what that is and we all need to know whether/how this intervention may or many not have impacted risk attitudes/calculation process/perception/etc.Discuss the potential for the effect to have been driven by an experimenter demand effect rather than learning what is optimal.  In the process you are implicitly telling them that what they did before was sub-optimal. Could things be different were participants to engage in the same exercise but without the heavy handed framing the design imposed?Given the broad audience at PLoSOne and connection of your paper and topics to the economics field I encourage you to provide some analysis more consistent with norms of economics papers as well as your current analysis that is more in line with Psych and Neuro papers.  Sign-tests of within subject Avg Risky Choice % Pre-post intervention in each AC and NC and Control.  Likewise with Avg Ambiguity Choice % pre-portMann-whitney or Wilcoxon tests of the the difference Pre-post compared between AC and Control,  NC and Control, and AC and NCGiven the rich multi-observation nature of the data a Fixed Effect Probit analysis of Risky and Ambiguity choice with dummies for the different treatments and other control variables would be nice in addition to the ANOVAs presented currently. 

We would appreciate receiving your revised manuscript by Nov 14 2019 11:59PM. To enhance the reproducibility of your results, we recommend that if applicable you deposit your laboratory protocols in protocols.io, where a protocol can be assigned its own identifier (DOI) such that it can be cited independently in the future. For instructions see: http://journals.plos.org/plosone/s/submission-guidelines#loc-laboratory-protocols

We look forward to receiving your revised manuscript.

Kind regards,

Jason Anthony Aimone

Academic Editor

PLOS ONE

**Journal Requirements:**

**Comments to the Author**

1. Is the manuscript technically sound, and do the data support the conclusions?

Reviewer #1: Yes

Reviewer #2: Yes

2. Has the statistical analysis been performed appropriately and rigorously? 

Reviewer #1: Yes

Reviewer #2: Yes

3. Have the authors made all data underlying the findings in their manuscript fully available?

Reviewer #1: Yes

Reviewer #2: Yes

4. Is the manuscript presented in an intelligible fashion and written in standard English?

Reviewer #1: Yes

Reviewer #2: Yes

5. Review Comments to the Author

Reviewer #1: I found the study to be interesting, and the experimental design and statistical analysis to be sound. I have the following minor comments:

1) In the raw data files, the ReadMe file indicates that the choice.RESP variable representing a subject's choice in a given trial should be either 0 or 1. However, some choice responses (e.g., for subject 1923 in the EF folder) takes values 0, 1, or 2. It would be great if the authors can clarify this discrepancy.

2) I would suggest that the authors cite Liu and Colman (2009), "Ambiguity aversion in the long run: Repeated decisions under risk and uncertainty" (Journal of Economic Psychology, 30(3): 277-284). That paper presents a different experiment of repeated choices under ambiguity without an intervention and finds that ambiguity aversion is reduced in repeated-choice environments relative to a single-choice condition.

Reviewer #2: This study examined the extent to which two different interventions reduced ambiguity aversion in an economic lottery decision-making task. This study definitely contributes to the growing literature on ambiguity and risk attitudes especially in laying foundational work on conducting interventions on reducing biases that lead to suboptimal decision-making. There are a few areas, however, in which the authors could provide more justification regarding their modeling procedure and details of the task. My comments are enumerated below.

1. The authors state in the introduction that many individuals “appear pessimistic in estimating the real outcome probability.” This seems to run contrary to a large body of literature in psychology showing that individuals exhibit an optimism bias especially when involving personal risks (see Neil Weinstein and Tali Sharot’s work). Do individuals tend to exhibit pessimistic behavior only in economic tasks? Second, the authors state on line 98, p.5 that risk and ambiguity attitudes are largely uncorrelated. However, even in one of the papers the authors cited, it states that risk and ambiguity attitudes are weakly correlated. Weak correlations are not the same as no correlations. In fact, the authors end up finding that risk and ambiguity attitudes are indeed correlated in both the pre- and post-intervention conditions.

2. The authors should provide more information regarding the details of experimentation. Did participants do the experiment on the computer? How long did the task take to complete? How long were they given to make a decision? Before the pre-intervention session, were the participants able to practice the task?

3. Did the authors control for individual differences in numeracy?

4. Did the authors examine reaction time to make a decision when choosing between the reference lottery and either an ambiguous or risky gamble? This would allow the authors to test whether participants were actually trying to do the calculations that they had been taught either in the AC or NC intervention by comparing pre and post decision RTs.

5. How many times did the participants in the AC condition receive guidance or feedback on the correct answer during computing objective probabilities? Was there variability in the extent to which participants were able to do the calculations? I know the samples were relatively small, but it would be interesting to know which people were most affected by the AC or NC intervention. For instance, are people who are extremely ambiguity averse or risk averse, make the most changes in their behavior.

6. Add to the limitations that this was a Yale University population. It is likely that this population may not be representative of the general population.

7. Typically in these ambiguity/risk lottery choice tasks, participants are usually required to choose between a certain option and a risky or ambiguous option. In the present study, the authors used a reference lottery (50% of $5) and either a risky or ambiguous option. What was the rationale for giving participants a reference lottery versus a certain option?

8. In the introduction the authors state that risk and ambiguity attitudes are “largely uncorrelated.” Then, why on pg. 11 lines 226-231 does ambiguity attitude rely on a person’s risk attitude? This implies that there is a relationship between ambiguity and risk attitudes.

9. Why did the authors fit each participant’s choices in the risky trials instead of estimating risk and ambiguity together in the same model as in Levy et al., 2010 and other studies using a similar task? Would the authors get similar results if they used a similar model to estimate risk and ambiguity attitudes simultaneously within the same model? Similarly, would the changes in ambiguity attitudes and risk attitudes show the same results as the metric of proportion of choices?

10. How did the authors come up with the limits for [0, 2.0987] for the risk attitude parameter? Was this based on simulations using this task design? If so, the authors should provide some information about this procedure for others to clarity and replication purposes.

11. How were the participants’ choices pre-processed? Please provide additional details, and what function in MATLAB was used to estimate risk and ambiguity attitudes? The authors provided the packages used in R but not in MATLAB.

12. The first paragraph under results was redundant text from the Methods section of the paper. I suggest the authors remove this text.

13. In S1, the authors state “what is your chance of winning across these two lotteries? And the participant put 50 out of 100. Why is this incorrect? As the next slide then says, your chance of winning in these lotteries is actually 50 out of 100.

14. I’m very unclear on how the subjects were given feedback regarding their responses, was it just the fact that there was the “return” button on the screen after they submitted their responses? Were they ever shown the actual answer or were they just expected to eventually reach the correct answer? Somewhere, in the supplementary material it would be helpful if the authors actually provided what the accurate response should be.

15. The authors should provide a rationale for why they only examined trials in which the ambiguous lottery offered more than $5, so this basically excludes trials in which subjects were presented risky/ambiguity lotteries of $5. Why exclude these? Is this a typical procedure for estimating ambiguity attitudes? It seems that the authors are selectively choosing the conditions to determine a person’s ambiguity attitude. How people behave when the ambiguous lottery offered the same amount as the reference lottery may also inform their ambiguity attitude. What are the results if the authors choose to include all trials?

16. Why do the authors think that there were no differences between the AC and NC condition? Given that many of their findings demonstrated the similarity between these two conditions, it would be helpful for the reader if the authors provided some explanations as to what factors may have contributed to these findings?

17. Did the authors collect any variables that could be indicators of SES? If so, the authors could potentially test the affects of SES on ambiguity and risk attitudes as suggested in their discussion.

6. PLOS authors have the option to publish the peer review history of their article (what does this mean?). If published, this will include your full peer review and any attached files.

Reviewer #1: No

Reviewer #2: No

---

## [Author Response · Author response to Decision Letter 0]

13 Dec 2019

Dear Dr. Aimone,

Thank you so much for the opportunity to revise our manuscript, entitled “Learning about the Ellsberg Paradox reduces, but does not abolish, ambiguity aversion”.

We have found your and the reviewers’ comments and suggestions extremely helpful. In the attached revised manuscript, we have addressed all of these points. We believe the manuscript is much improved as a result of this revision, and hope it is now suitable for publication in PLoSOne.

Sincerely,

Ruonan Jia and Ifat Levy

 

We thank the editor and the reviewers for their supportive reviews and helpful comments. Below we provide a point-by-point response to comments. Line number indicates the location in the track-change version of the manuscript when all mark-ups are shown.

I made the decision on your manuscript after receiving comments from two scholars in the field and giving the paper a review myself. Should you decide to submit a revision of your paper please be sure to address all the comments by the reviewers in your response document. I would like you to in particular address the data issue raised by reviewer #1. In addition to Rev1 and Rev2's comments please incorporate responses to a few things from me as well.

Define within the text what your "Base rate neglect" intervention taught. Many readers will not know what that is and we all need to know whether/how this intervention may or many not have impacted risk attitudes/calculation process/perception/etc.

Sorry for our oversight on this. We added a brief description of the “base rate neglect” intervention in the “Experimental procedure” section as follows: 

Briefly, this intervention taught participants that base rate (the unconditional frequency of an event) should be taken into account when calculating the true probability of a specific conditional event. This was conveyed through the “blue-green cab problem”: suppose a witness reported seeing a blue cab causing a hit-and-run accident; what is the probability of the witness correctly identifying the hit-and-run cab as blue, given (1) the chance of the witness correctly identifying the color of the cab, and (2) the ratio of blue and green cabs (base rate) in the city. Similar to the AC intervention, participants were guided in performing step-by-step calculations, and reached the correct answer through feedbacks on incorrect answers. This intervention required participants to calculate the conditional probability, but was not related to ambiguity (or to value-based choice in general), and thus could serve as a control. (Line 217-226)

Readers can infer from this brief description, without turning to the supplementary, that this intervention involves mathematical calculation, but no risk or ambiguity, and thus should not change participants’ behavior in principle. 

Discuss the potential for the effect to have been driven by an experimenter demand effect rather than learning what is optimal. In the process you are implicitly telling them that what they did before was sub-optimal. Could things be different were participants to engage in the same exercise but without the heavy handed framing the design imposed?

Thank you for this suggestion. We added the discussion of potential experimenter demand effect in the “Limitation of current study and directions for future research” section in the discussion, as follows:

Another potential limitation is that our interventions implicitly informed participants that their behavior had been suboptimal, which may have led to an experimenter demand effect [(65)], rather than true behavioral change through learning. Although we cannot completely rule out such an effect, we note that participants did seem to understand the Ellsberg paradox (Fig 2), suggesting that the behavioral change could not be fully driven by experimenter demand. In future attempts to replicate the effect, it will be interesting to use an intervention which walks participants through probability calculations without revealing the best strategy. (Line 709-715)

Given the broad audience at PLoSOne and connection of your paper and topics to the economics field I encourage you to provide some analysis more consistent with norms of economics papers as well as your current analysis that is more in line with Psych and Neuro papers. 

Thank you for this suggestion – we have added these analyses, as detailed below.

Sign-tests of within subject Avg Risky Choice % Pre-post intervention in each AC and NC and Control. Likewise with Avg Ambiguity Choice % pre-port

We added the paired Sign tests of pre-post ambiguous choice proportion (Line 416), risk-corrected ambiguity attitude (Line 440), and risky choice proportion (Line 510) within each group. The results were consistent with our existing analysis, except that the change in risky choice proportion was not significant after correction for multiple comparisons across the three intervention groups. This suggests that the intervention effect on risky choice was weak, although we note that this effect was significantly higher in the AC and NC groups compared to the control group as revealed by the ANOVA.

Mann-whitney or Wilcoxon tests of the the difference Pre-post compared between AC and Control, NC and Control, and AC and NC

We added the Mann-Whitney U tests comparing pre-post difference of ambiguous choice proportion (Line 426), risk-corrected ambiguity attitude (Line 454), and risky choice proportion (Line 521) between each pair of the intervention groups. The results were consistent with our existing analysis, that the intervention effect was significant in AC and NC groups compared with control, but not different between AC and NC.

Given the rich multi-observation nature of the data a Fixed Effect Probit analysis of Risky and Ambiguity choice with dummies for the different treatments and other control variables would be nice in addition to the ANOVAs presented currently. 

We conducted a mixed-effect logistic regression, to investigate the intervention’s effect on choices of the ambiguous and risky lotteries separately, controlling for other factors and individual random effects. Details about the methods were added to the “Data analysis: Mixed-effect generalized linear model (GLM) approach” section (Line 317), as follows:

In addition to the model-based and model-free uncertainty attitudes estimation, we also conducted a mixed-effect logistic regression of choices to control for other factors and individual random effect more rigorously. We modeled the choice of the varying lottery in ambiguous and risky trials separately, using generalized linear models, with logit as the link function (equations 3 and 4). 

Ambiguous trials: choice ~ phase + intervention + phase × intervention + ambiguity level + value + EP score + EP score × intervention + age + gender + (1 + phase | id), with logit as the link function. (3)

Risky trials: choice ~ phase + intervention + phase × intervention + risk level + value + EP score + EP score × intervention + age + gender + (1 + phase | id), with logit as the link function. (4)

In addition to the primary focus of the effects of phase (pre- or post- intervention; within-subject), intervention (AC, NC, and control; between-subject), and the interaction between the two, we included the ambiguity level (only for ambiguous trials), risk level (only for risky trials), and lottery outcomes (value) as the lottery design factors that influence choices. We also included the score of understanding the Ellsberg Paradox (EP score), the interaction between this score and intervention, age, and gender as other controlling factors, and individual (id) as the random effect. Other than categorical variables including phase, intervention method, gender, and participant id, all variables were standardized before fitting the GLM. For the same reason as stated above, we excluded trials with the varying lottery offering $5 in this analysis as well. 113 participants with a score of understanding the Ellsberg Paradox were included in this analysis.

We presented the results in Line 480 for ambiguous trials and Line 526 for risky trials. They were consistent with our previous ANOVA tests.

Reviewer #1: I found the study to be interesting, and the experimental design and statistical analysis to be sound. I have the following minor comments:

1) In the raw data files, the ReadMe file indicates that the choice.RESP variable representing a subject's choice in a given trial should be either 0 or 1. However, some choice responses (e.g., for subject 1923 in the EF folder) takes values 0, 1, or 2. It would be great if the authors can clarify this discrepancy.

Thank you for spotting this insufficient explanation. A value of 2 in the choice response column indicates missing response. We have added this explanation in the ReadMe file in the S9 supplementary material folder. We also added more clarification in the ReadMe file, and uploaded a sheet of processed data and statistical analysis scripts in R to the S9 folder, in addition to the raw participant choice data.

2) I would suggest that the authors cite Liu and Colman (2009), "Ambiguity aversion in the long run: Repeated decisions under risk and uncertainty" (Journal of Economic Psychology, 30(3): 277-284). That paper presents a different experiment of repeated choices under ambiguity without an intervention and finds that ambiguity aversion is reduced in repeated-choice environments relative to a single-choice condition.

Thank you for bringing up this paper. We added this discussion to the section “Ambiguity aversion was reduced following the intervention” in the discussion, as follows:

Providing outcome feedback for each and every choice, rather than for a single choice as we did here, can also reduce ambiguity aversion [(48)] (Line 626).

Reviewer #2: This study examined the extent to which two different interventions reduced ambiguity aversion in an economic lottery decision-making task. This study definitely contributes to the growing literature on ambiguity and risk attitudes especially in laying foundational work on conducting interventions on reducing biases that lead to suboptimal decision-making. There are a few areas, however, in which the authors could provide more justification regarding their modeling procedure and details of the task. My comments are enumerated below.

1. The authors state in the introduction that many individuals “appear pessimistic in estimating the real outcome probability.” This seems to run contrary to a large body of literature in psychology showing that individuals exhibit an optimism bias especially when involving personal risks (see Neil Weinstein and Tali Sharot’s work). Do individuals tend to exhibit pessimistic behavior only in economic tasks? Second, the authors state on line 98, p.5 that risk and ambiguity attitudes are largely uncorrelated. However, even in one of the papers the authors cited, it states that risk and ambiguity attitudes are weakly correlated. Weak correlations are not the same as no correlations. In fact, the authors end up finding that risk and ambiguity attitudes are indeed correlated in both the pre- and post-intervention conditions.

Thank you so much for pointing out these contradictions, we completely agree with both comments and have revised the paper accordingly. 

First, the reviewer makes a great point about the apparent contradiction between the widely observed optimism bias, and interpretation of ambiguity aversion (also widely observed) as pessimism. We have changed our terminology in the introduction into “Many individuals exhibit aversion to ambiguity about outcome likelihoods, especially in the realm of economic decision making”, and touch upon the potential relationship between ambiguity attitudes and optimism bias in the discussion, as follows:

Another interesting question is whether and how ambiguity aversion relates to the “optimism bias”, which is commonly reported, where people actually overestimate the likelihood for positive future events (such as having a long life expectancy), and underestimate the likelihood for negative events (such as getting divorced) [(63), (64)]. (Line 673).

Second, we apologize for inaccurately presenting the relationship between risk and ambiguity attitudes, and have changed the wording to “not strongly correlated” (Line 100). Generally, correlation between ambiguity and risk attitudes was not found in some studies (e.g. Levy et al., 2010, Huettel et al, 2006), and was weak in other studies (e.g. Tymula et al., 2013). In the current study, we found significant but not strong correlation (pre-intervention, Pearson’s r = 0.49; post-intervention, Pearson’s r = 0.29). 

2. The authors should provide more information regarding the details of experimentation. Did participants do the experiment on the computer? How long did the task take to complete? How long were they given to make a decision? Before the pre-intervention session, were the participants able to practice the task?

We apologize for our lack of clarity on this. Yes, participants completed the task on a computer, and it took about 45 minutes including study introduction. They were given a limit of 6 seconds to make a decision, and they practiced 6 trials before the pre-intervention choices. We added these details to the sub-sections of Trial structure (Line 160), Experimental procedure (Line 192), in the Method section. We also added the procedure diagram for the whole experiment, as panel E in Figure 1.

3. Did the authors control for individual differences in numeracy?

We did not measure numeracy skills in this study. Though we agree that it would be very interesting to look at the relationship between learning and individual differences in numeracy, we do not think it is a crucial factor in our task. Although understanding the Ellsberg paradox requires mathematical calculation, once participants accept the optimal decision strategy (to always choose the ambiguous lottery) that we taught them, they should be able to choose without actually calculating anything. Also, we note that in previous studies with similar paradigms, risk and ambiguity attitudes were not associated with numeracy skills in much more diverse samples (Tymula et al., 2013, Ruderman et al., 2016). 

4. Did the authors examine reaction time to make a decision when choosing between the reference lottery and either an ambiguous or risky gamble? This would allow the authors to test whether participants were actually trying to do the calculations that they had been taught either in the AC or NC intervention by comparing pre and post decision RTs.

We examined the reaction time, and found that (1) participants were generally weakly faster in ambiguous trials than risky trials in both pre- and post- intervention choices, regardless of intervention methods, and (2) participants were faster following the intervention, regardless of risky of ambiguous trials, and also regardless of intervention method. The first effect indicates that participants are not necessarily performing more complex calculation of the more complex ambiguous lotteries. They were actually a little faster in ambiguous trials, probably due to aversion to ambiguity. Since this effect exists in both pre- and post- intervention choices, and did not differ across intervention methods, we think it is a general indication that decision making under risk and ambiguity might involve different cognitive processes. The second effect could be explained by familiarity with the task, as the pre- and post-intervention trials were the same, except for random trial order. We included the analysis and figures in S8. We believe that these results suggest that RT could not reflect more information about the intervention effect in our data.

5. How many times did the participants in the AC condition receive guidance or feedback on the correct answer during computing objective probabilities? Was there variability in the extent to which participants were able to do the calculations? I know the samples were relatively small, but it would be interesting to know which people were most affected by the AC or NC intervention. For instance, are people who are extremely ambiguity averse or risk averse, make the most changes in their behavior.

Participants had to answer 5 questions during the AC intervention, and for the first four questions (guess what is the proportion of red-blue chips in the bag), they could try twice to get an answer within the correct range. If they were not correct in the second attempt and provided a number out of the ambiguity range, the computer would choose a number for them. For the last question (calculate the reward probability across both ambiguous lotteries with red and blue corresponding to the winning color), if they gave the wrong answer, they were given a hint before answering again. In the NC intervention, these calculation steps were just shown to the participants, and they were not stated as questions.

Unfortunately, we did not record participants responses during the intervention, and could not examine the variability of errors during calculation. However, we did look at the correlation between pre-intervention ambiguity attitude and the change in ambiguity attitude, and found that participants who were more ambiguity averse in pre-intervention choices changed their behavior the more. We added the results in the “Pre-intervention ambiguity attitude predicted the change in ambiguity aversion” result section as follows:

If learning the Ellsberg paradox successfully decreased participants’ ambiguity aversion, those who were most averse to ambiguity before the intervention should be most likely to increase their tolerance to ambiguity. To examine if this was the case, we looked at the correlation between pre-intervention ambiguity attitude (risk-corrected choice proportion in ambiguous trials) and the change in ambiguity attitude, combining participants from the AC and NC interventions. We observed a negative correlation between the two, such that participants who were most averse to ambiguity increased their ambiguity tolerance more (Fig 6A; Pearson’s r = -0.53, p < 0.001). Note, however, that this negative correlation could result from a ceiling effect, as participants who were not ambiguity averse before the intervention could not reduce their ambiguity aversion any further. We did not see such relationship between pre-intervention risk attitude and change in risk attitude (Fig 6B; Pearson’s r = -0.064, p = 0.572). (Line 558 and Figure 6). 

However, this result should be interpreted carefully. Because the upper limit of the choice proportion of the ambiguous lottery is one, the correlation could result from a ceiling effect: participants who were least ambiguity averse in pre-intervention choice could not reduce their ambiguity aversion any further. No correlation was found between pre-intervention risk attitude and change in risk attitude in AC and NC groups. 

6. Add to the limitations that this was a Yale University population. It is likely that this population may not be representative of the general population.

We agree, and added this limitation to the discussion “Limitation of current intervention methods” section as follows:

Our participants came from a Yale-New Haven community, among which many were Yale students or staff members. Social economic status was quite homogeneous, and was not representative of the general population. Future studies should investigate the effect of learning about the Ellsberg paradox in a broader and more diverse population. (Line 705).

7. Typically in these ambiguity/risk lottery choice tasks, participants are usually required to choose between a certain option and a risky or ambiguous option. In the present study, the authors used a reference lottery (50% of $5) and either a risky or ambiguous option. What was the rationale for giving participants a reference lottery versus a certain option?

We designed the task this way because it mimicked the Ellsberg paradox set-up. Since the objective winning probability of the ambiguous lottery is always 50% - same as the reference risky 50% lottery - participants who understood the Ellsberg paradox only needed to compare the amounts of the lottery payoff to make a choice. This design could let us investigate the intervention’s effect by directly looking at the choices in ambiguous trials. We now clarify this in the “Method: Payment mechanism” section as follows:

Importantly, this design ensured that the objective winning probability of all of the ambiguous lotteries was 50%, the same as the reference risky lottery. To understand why, …… Thus, regardless of the ambiguity level and of the actual distribution of blue and red chips in the bag, the winning probability of all the ambiguous lotteries was 50%. A participant aiming to maximize her earnings should therefore always choose the ambiguous lottery if it offers more than $5. Choosing the reference lottery (50% chance of $5) over an ambiguous lottery that offers more than $5 would indicate ambiguity aversion. (Line 176). 

Setting the risky 50% lottery as the reference was also used in previous studies (e.g. Levy et al. 2010) for estimating risk and ambiguity attitudes through behavioral models.

8. In the introduction the authors state that risk and ambiguity attitudes are “largely uncorrelated.” Then, why on pg. 11 lines 226-231 does ambiguity attitude rely on a person’s risk attitude? This implies that there is a relationship between ambiguity and risk attitudes. 

We apologize for the confusion – what we meant was not that the individual’s ambiguity attitude relies on her risk attitude, but rather that decision-making under ambiguity relies both on the individual’s ambiguity attitude and on her risk attitude. This is because under ambiguity the decision maker has to first assume some probability (relying on her ambiguity attitudes), and then apply her risk attitude to the assumed probability to make a choice. We now clarify that in the text.

9. Why did the authors fit each participant’s choices in the risky trials instead of estimating risk and ambiguity together in the same model as in Levy et al., 2010 and other studies using a similar task? Would the authors get similar results if they used a similar model to estimate risk and ambiguity attitudes simultaneously within the same model? Similarly, would the changes in ambiguity attitudes and risk attitudes show the same results as the metric of proportion of choices? 

Thank you for this question. Our intent was indeed to use the same model we have used in Levy et al 2010. However, for such model fitting, there must be sufficient variance in the choice data (i.e. participants should choose the varying lottery on some trials and the reference lottery on other trials). Following the intervention, many participants did not choose the ambiguous lottery as often, and several participants did not choose the ambiguous lottery at all. Thus, model estimates of ambiguity attitudes were not reliable and data from too many (n=30) participants had to be excluded. To overcome this problem we only fitted risky trials (which still included sufficient variance) and corrected the ambiguous choices using the fitted risk parameter. Furthermore, as mentioned above, because our design mimicked the Ellsberg paradox set-up, looking at choice data (and the risk attitude-corrected choice data) in ambiguous trials without model fitting is good enough for answering our research question. 

10. How did the authors come up with the limits for [0, 2.0987] for the risk attitude parameter? Was this based on simulations using this task design? If so, the authors should provide some information about this procedure for others to clarity and replication purposes.

We added the rationale and the detailed explanation of how we came up with the constraints, in the Data analysis: Estimation of risk and ambiguity attitudes section as follows:

The lower boundary was determined by equating the subjective value of the best lottery (38% chance of $65) with the subjective value of the reference lottery (50% chance of $5), using equation 1. Similarly, to determine the upper boundary, we equated the subjective values of the worst lottery (13% chance of $9.5) and the reference lottery (50% chance of getting $5). (Line 298). 

We also apologize for the typo - the lower limit is actually 0.1070, not 0. We assure the reviewers that it is only a typo, and the actual fitting was indeed constrained between [0.1070, 2.0987] (we include the model-fitted data in S9). 

The constraints are based on the theoretical range of risk attitudes that our design can reveal. To reach an estimate, we need the participant to choose the varied lottery on some trials, and the reference lottery on other trials. To understand why, let’s assume the participant has chosen the variable lottery on all trials. This tells us that for this participant, the subjective values of all the variable lotteries were higher than (or at least equal to) the subjective value of the reference lottery, which means this participant is risk seeking. But how risk seeking? There is no way of knowing. It could be that the subjective value of the lowest varied lottery (13% of $9.5) was just equal to the subjective value of the reference lottery (50% of $5; slight risk seeking). But it could also be that the subjective value of the lowest varied lottery was much higher – this would not affect the choice pattern, as in either case the participant is only choosing the varied lottery. Thus, the upper limit of risk seeking we can detect is when the subjective value of [13% of $9.5] is equal to the subjective value of [50% of $5]. By the same logic, if a participant only chose the reference lottery, we know that she is risk averse, and the limit of risk aversion we can detect is when the subjective value of the best varied lottery (38% of $65) is just equal to that of the reference lottery. Using Equation 1 in the main text, we calculated the subjective values for these cases and computed the upper and lower alphas that we can detect with our design. Note that the same approach has been used before (Ruderman et al. 2016).

11. How were the participants’ choices pre-processed? Please provide additional details, and what function in MATLAB was used to estimate risk and ambiguity attitudes? The authors provided the packages used in R but not in MATLAB.

We added the MATLAB function used for model fitting in the “Statistical analysis” section, as follows:

Participants’ choice data were pre-processed to get the choice proportion of risky and ambiguous lotteries, and model-fitted in MATLAB (Version R2016b, MathWorks) to obtain the model-based ambiguity attitudes (see preprocessing and model-fitting scripts in S9). The procedure of maximizing log likelihood (minimizing negative log likelihood) was implemented through the MATLAB function fmincon, from the Optimization Toolbox, with the aforementioned α constraint. (Line 342)

12. The first paragraph under results was redundant text from the Methods section of the paper. I suggest the authors remove this text.

We deleted the text.

13. In S1, the authors state “what is your chance of winning across these two lotteries? And the participant put 50 out of 100. Why is this incorrect? As the next slide then says, your chance of winning in these lotteries is actually 50 out of 100.

Thank you for pointing out this confusion. The yellow input of 50 on this page is to show the correct response. We have added clarification on the second page, to avoid more this confusion.

The “INCORRECT” button was not part of the experiment, but is there for the reader to click on, so they would see what the next slide would be IF the participant submitted an incorrect response. After clicking on the “INCORRECT” button, the reader will be led to the slide which says “Remember, you estimated your chance of winning when……” and the participant was required to answer the same question again with this hint. Then the reader could click on “RETURN” button on the right corner, and move on to the next slide as would be shown in the experiment. 

We now explain at the beginning of the demonstration how these “INCORRECT” buttons work, and hope that readers will be able to interactively experience the intervention as similar as possible to the actual experiment run through E-prime on the computer.

14. I’m very unclear on how the subjects were given feedback regarding their responses, was it just the fact that there was the “return” button on the screen after they submitted their responses? Were they ever shown the actual answer or were they just expected to eventually reach the correct answer? Somewhere, in the supplementary material it would be helpful if the authors actually provided what the accurate response should be.

Again, we apologize for the confusion on this. The “INCORRECT” and “RETURN” buttons did not appear in the actual intervention, we now clarify this in the supplementary material S1.

To your question, when they made a mistake, participants were required to submit the answer again. In the first four questions, where participants were asked how many blue or red chips were in the bag and each lottery’s winning probability, they were given two chances to submit answers. If they still did not get it right, the program would choose an answer for them. The reader can see this by clicking the “INCORRECT” button twice, after which there will no longer be an “INCORRECT” button. In the fifth question, they were asked what was the chance of winning across two lotteries. If they did not answer correctly it in the first attempt, they received a hint, and were required to answer it again. 

15. The authors should provide a rationale for why they only examined trials in which the ambiguous lottery offered more than $5, so this basically excludes trials in which subjects were presented risky/ambiguity lotteries of $5. Why exclude these? Is this a typical procedure for estimating ambiguity attitudes? It seems that the authors are selectively choosing the conditions to determine a person’s ambiguity attitude. How people behave when the ambiguous lottery offered the same amount as the reference lottery may also inform their ambiguity attitude. What are the results if the authors choose to include all trials?

This is a good question - we added a more detailed explanation of the rationale for excluding the $5 trials in calculating ambiguity attitudes, in the “Estimation of risk and ambiguity attitudes” section as follows:

Here too we did not include trials in which the ambiguous lottery offered $5, because in such trials participants should not necessarily choose the ambiguous option, even if they have completely abolished ambiguity aversion. To make sure that this approach did not bias the results, however, we also repeated the analysis with the $5 trials included (see S2). (Line 268).

We chose to exclude these trials in order to cleanly estimate ambiguity attitude and make the result more interpretable. At the end of both the AC and NC interventions, we explicitly told participants that because the theoretical winning probability of the ambiguous lottery was the same as the reference risky lottery (50%), they should always choose the ambiguous lottery which offered more money. This is not the case, however, in trials in which both the ambiguous and reference lotteries offered $5. In principle, after the intervention, if participants fully abolished ambiguity aversion, they should be indifferent between these lotteries, and should choose the ambiguous lottery 50% of the time. So including $5 trials in the analysis would result in ambiguous choice proportion that is lower than 1, even for a completely ambiguity-neutral participant, making the results less interpretable. 

Having said that, because of the reviewer’s concern that this exclusion would bias the results, we also calculated the choice proportion of the ambiguous lotteries including $5 trials, and now present the results in supplementary material S2. Both approaches lead to similar overall results.

16. Why do the authors think that there were no differences between the AC and NC condition? Given that many of their findings demonstrated the similarity between these two conditions, it would be helpful for the reader if the authors provided some explanations as to what factors may have contributed to these findings?

Similar to the reviewer, we have also expected to see a difference in learning between the two conditions. We believe that the lack of difference is due to (1) the limitation of Yale-New Haven highly educated sample, and (2) the math involved in the Ellsberg paradox is simple and the best decision strategy is straightforward.

17. Did the authors collect any variables that could be indicators of SES? If so, the authors could potentially test the affects of SES on ambiguity and risk attitudes as suggested in their discussion.

We only collected age, education, and job status from the participants. As we mentioned, this is a Yale-New Haven sample, and many participants were Yale students or staff members. Thus, our sample does not have enough variance in terms of SES. We completely agree that it would be interesting to collect SES data from a more diverse sample, and investigate the effect as we suggested in the discussion “Future studies should investigate the effect of learning about the Ellsberg paradox in a broader and more diverse population” (Line 707).

---

## [Decision Letter · Decision Letter 1]

24 Jan 2020

Learning about the Ellsberg Paradox reduces, but does not abolish, ambiguity aversion

PONE-D-19-23100R1

Dear Dr. Jia,

We are pleased to inform you that your manuscript has been judged scientifically suitable for publication and will be formally accepted for publication once it complies with all outstanding technical requirements.

With kind regards,

Jason Anthony Aimone

Academic Editor

PLOS ONE

Additional Editor Comments (optional):

Reviewers' comments:

Reviewer's Responses to Questions

**Comments to the Author**

1. If the authors have adequately addressed your comments raised in a previous round of review and you feel that this manuscript is now acceptable for publication, you may indicate that here to bypass the “Comments to the Author” section, enter your conflict of interest statement in the “Confidential to Editor” section, and submit your "Accept" recommendation.

Reviewer #1: All comments have been addressed

Reviewer #2: All comments have been addressed

2. Is the manuscript technically sound, and do the data support the conclusions?

Reviewer #1: (No Response)

Reviewer #2: Yes

3. Has the statistical analysis been performed appropriately and rigorously? 

Reviewer #1: (No Response)

Reviewer #2: Yes

4. Have the authors made all data underlying the findings in their manuscript fully available?

Reviewer #1: (No Response)

Reviewer #2: Yes

5. Is the manuscript presented in an intelligible fashion and written in standard English?

Reviewer #1: (No Response)

Reviewer #2: Yes

6. Review Comments to the Author

Reviewer #1: (No Response)

Reviewer #2: (No Response)

7. PLOS authors have the option to publish the peer review history of their article (what does this mean?). If published, this will include your full peer review and any attached files.

Reviewer #1: No

Reviewer #2: Yes: Nina Lauharatanahirun

---

## [Editor Report · Acceptance letter]

7 Feb 2020

PONE-D-19-23100R1 

Learning about the Ellsberg Paradox reduces, but does not abolish, ambiguity aversion 

Dear Dr. Jia:

I am pleased to inform you that your manuscript has been deemed suitable for publication in PLOS ONE. Congratulations! Your manuscript is now with our production department. 

With kind regards,

on behalf of

Dr. Jason Anthony Aimone 

Academic Editor

PLOS ONE